# ERiC-UP³ Benchmark: E-Commerce Risk intelligence Classifier for Detecting Infringements on Utility Patent and Product Pairs

## Abstract

Innovation is a key driver of economic and social progress, with Intellectual Property (IP) protection through patents playing a crucial role in safeguarding new creations. For businesses actively producing goods, detecting potential patent infringement is vital to avoid costly litigation and operational disruptions. However, the significant domain gap between products and patents—coupled with the vast scale of existing patent databases—makes infringement detection a complex and challenging task. Besides, the machine learning (ML) community has not widely addressed this problem, partly due to the lack of comprehensive datasets tailored for this task. In this paper, we firstly formulate a new task: detecting potentially infringing patents for a given product represented by multi-modal data, including images and textual descriptions. This task requires a deep understanding of both technical and legal contexts, extending beyond simple text or image matching to assess functional similarities that may not be immediately apparent. To promote research in this challenging area, we further introduce the ERiC-UP³ (**E**-Commerce **R**isk **i**ntelligence **C**lassifier on **U**tility **P**atent and **P**roduct **P**airs) benchmark, a large-scale, well-structured dataset comprising over 13-million patent samples and 1 million product samples. It includes 11,000 meticulously annotated infringement pairs for training and 2,000 for testing, all rigorously reviewed by patent experts to ensure high-quality annotations. The dataset reflects real-world scenarios with its multi-modal nature and the necessity for deep functional understanding, offering unique characteristics that set it apart from existing resources. As a case study, we provide results from a series of baseline methods and propose a simple yet effective infringement detection pipeline. We also explore additional approaches that may enhance detection performance, such as text style rewriting, cross-modal matching effectiveness, and image domain alignment. Overall, the ERiC-UP³ benchmark is the first strictly annotated product-patent infringement detection dataset and stands as the largest multi-modal patent dataset, as well as one of the largest multi-modal product datasets available. We aim to advance research extending language and multi-modal models to diverse and dynamic real-world data distributions, fostering innovation and practical solutions in IP infringement detection.

## 1 Introduction

Intellectual property (IP) protection through patents is essential for safeguarding innovations across industries, granting companies and individuals exclusive rights over their creations (Reitzig & Puranam, 2009; Maskus, 1998). However, the misuse of the patent system—notably by "patent trolls" who file lawsuits for compensation without ever producing the patented products—places significant legal and financial burdens on legitimate manufacturers (Golden, 2006). For businesses actively producing goods, avoiding IP infringement is critical to prevent costly litigation and operational disruptions. Proactively detecting and mitigating potential infringements is key to minimizing these risks, ensuring smoother operations, and fostering continued innovation.

A major challenge in detecting patent infringement lies in the *significant domain gap* between products and patents. As illustrated in Figure 1, patent documents contain technical text and schematic

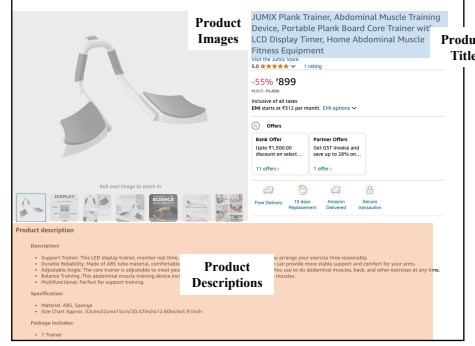

Figure 1: Two pages of an example patent document (left, *Plank support exercise apparatus and related methods*, Publication No.US10286245B2) and website of an example product (right, *planks core trainer abdominal board lcd display strength training fitness*). The highlighted sections show subsets of data fields that mainly function and we put zoomed-in example in in Appendix A.

diagrams, while product data typically consist of images and textual descriptions. This multi-modal nature results in stark differences in both visual and textual representations, complicating cross-modal matching and functional similarity assessment. Moreover, patents often describe abstract ideas, technical processes, and innovative designs that require *a deep understanding of both legal and technical contexts*. On the other hand, the vast number of existing patents—tens of millions—further exacerbates the difficulty, creating an *immense search space* where accurately retrieving relevant patents for a given product becomes a daunting task. Beyond the sheer volume, the ambiguity in language and variation in the way inventions are described add layers of complexity. Functional overlap between products and patents may not be immediately apparent from surface-level similarities, meaning infringement may occur based on underlying mechanisms rather than visible features, which necessitates deeper analysis beyond simple text or image matching.

Usually, patent infringement detection has been a labor-intensive process requiring substantial time and expertise from specialists well-versed in both legal and technical fields (Bergmann et al., 2008). This reliance on manual assessments not only makes the process costly but also limits its accessibility, particularly for small and medium-sized enterprises. While leveraging ML (Goodfellow et al., 2016) to automate this process presents a promising solution to reduce costs and enhance accessibility, the absence of well-annotated datasets specifically designed tailored for this task has significantly impeded progress. Therefore, the field has not yet attracted widespread attention within the ML community, hindering the development of accurate and functional automated detection models.

This paper focuses on bridging this gap and advancing related research, where we first formulate the task of patent infringement detection and subsequently introduce two benchmark datasets tailored for it: ERiC-UP$^3$-Base and ERiC-UP$^3$-Large (**E**-Commerce **R**isk **i**ntelligence **C**lassifier on **U**tility **P**atent and **P**roduct **P**airs). The former includes a smaller set of samples designed to provide researchers with a platform for rapid testing and prototype development, while the latter offers broader data coverage, suitable for testing model robustness and effectiveness in more complex and diverse real-world scenarios. On these benchmarks, we conduct extensive experiments and provide a series of baseline results to demonstrate the task's challenges and to serve as a reference for subsequent research. Additionally, we develop a straightforward yet effective text-based method for infringement detection, which includes a classifier for potential patent infringements and a product-patent-specific retriever. Crucially, our framework shows strong baseline results and significantly enhances the performance of infringement detection. We further conduct a series of meaningful experiments on this dataset to improve the success rate of infringement detection, including text style rewriting, the incorporation of image knowledge to aid detection performance, and the alignment of image modal features. Through these contributions, we not only advance the application of ML in the field of IP protection but also provide valuable resources for the research community.

## 2 ERiC-UP$^3$ Benchmark

### 2.1 Task Formulation

A product item $P_i = (I_i^P, T_i^P)$ consists of a set of images $I_i^P = \{I_{i,1}^P, I_{i,2}^P, \cdots\}$ and a corresponding textual structure $T_i^P$, which includes a title and a description. Given a *gallery* set of patent samples

Table 1: Statistics of ERiC-UP³-Base and ERiC-UP³-Large.

| | Support Product Set | Gallery Set of Patents | Patent CPC Main Classes | Training Paris | Test Parins |
|---|---|---|---|---|---|
| Base | 979,438 | 2,551,842 | 5 | 7.349 | 454 |
| Large | | 13,410,443 | 137 | 11,000 | 2,000 |

$Q = \{Q_j | Q_j = (I_j^Q, T_j^Q)\}$, where each patent $Q_j$ includes a set of images $I_j^Q = \{I_{j,1}^Q, I_{j,2}^Q, \dots\}$ and a corresponding complex textual structure $T_j^Q$, typically comprising a title, abstract, background, claims, and other sections, the task is to retrieve the most functionally similar patent $Q_k \in Q$ that may be infringed upon by the query product item $P_i$. For example, the goal is to predict a ranked list $R_i = [id_i^1, id_i^2, \dots, id_i^k, \dots, id_i^N] \quad \forall id_j \in Q$, where $N$ indicates the size of querying patent pool and $id_i^k$ corresponds to a specific patent in $Q$, ordered by their relevance or likelihood of infringement with respect to the product item $P_i$. The task is to ensure that the most functionally similar and potentially infringing patent is optimally ranked highest on the list.

## 2.2 DATASET CONSTRUCTION AND SIZE

As shown in Table 1, we initially introduce ERiC-UP³-Large benchmark, which includes 11,000 pairs of product-patent infringements for training and 2,000 pairs for test. The large version also includes a gallery set of 13-million patents, complete with technical texts and diagrams. To facilitate fast validation of algorithm development while reducing training costs, we create ERiC-UP³-Base as a subset of the Large version. The Base version comprises 7,300 training pairs, 454 test pairs, and a reduced patent retrieval pool of 2.55 million patents, focusing on five specific patent CPC[1] categories (i.e., A45, A47, A63, B65 and H01). Each infringement pair in both versions is meticulously labeled by patent experts through three rounds of cross-validation, ensuring the identification of clear infringement cases. In addition, we collect of 1M multi-modal product samples, designed to *support* effective product representation learning through diverse image and textual descriptions. All patent and product samples are collected from the US Patent and Amazon websites[2]. The process of obtaining these samples, standardizing data formats, filtering out missing and erroneous entries, deduplicating, and merging the datasets into a user-friendly format is nontrivial. We highlight the several significant differences between our benchmark with previous related datasets in Appendix B.

## 2.3 CHALLENGE OF LABELING PRODUCT-PATENT INFRINGEMENT PAIRS

Labeling product-patent infringement pairs is a complex and demanding task due to the inherent difficulties in accurately linking products to the patents they may infringe upon. One of the primary challenges stems from the use of Virtual Patent Marking (VPM) (Patent & Office, 2014), a method mandated by US law requiring companies to disclose product-patent information. However, compliance is minimal, and the data provided is often sparse and unevenly distributed across industries. Manufacturers frequently list only product model numbers without detailed descriptions, making it arduous to ascertain the exact product specifications and associated patents. Patent experts in our IP team have dedicated considerable effort to meticulously verify and establish the relationships between these model numbers, the corresponding products, and their relevant patents. In addition to VPM data, a significant portion of our infringement pairs has been compiled through the diligent work of our IP team during pre-listing IP audits. Their deep understanding of patent databases and manual search methodologies enables them to identify and match patents with high precision. Furthermore, we have incorporated data from historical infringement cases, reflecting a wealth of knowledge accumulated over time. These comprehensive efforts underscore not only the challenges in assembling accurate product-patent pairs but also highlight the critical role of specialized expertise in navigating the intricacies of IP law and enforcement.

## 2.4 DATASET PRE-PROCESSING AND ATTRIBUTE

For data preprocessing, we first tackle textual data by structuring it. Each section of the text is parsed and de-duplicated individually, irrelevant characters are filtered using the duplicated n-gram coverage ratio (Rae et al., 2021), and the cleaned text is stored in a structured JSON format. For images, both product and patent datasets contain substantial noise, such as partial product images

---

[1]IPC was established by the 1971 Strasbourg Agreement. CPC, an extension of IPC, has been used by the USPTO since 2013, offering broader coverage and including a "Y" section for newer technologies.

[2]As specified by US law and Amazon, all patent and product data is publicly accessible.

Table 2: Statistics of textual sections.

| | Section | ERiC-UP$^3$-Base Avg # Words | ERiC-UP$^3$-Large Avg # Words |
|---|---|---|---|
| Gallery set of Patent | Title | 7.6 | 7.77 |
| | Abstract | 111.01 | 104.28 |
| | Claims | 894.48 | 946.61 |
| | Background | 99.65 | 96.83 |
| | CPC Code | - | - |
| | Publication Number | - | - |
| | Publication Month | - | - |
| | #imgs / sample | 21.01 | 20.51 |
| Support set of Product | Title | 11.72 | |
| | Description | 122.83 | |
| | #drawings / sample | 10.77 | |

Table 3: Comparison with previous datasets.

| Dataset | #Samples | Modality | Domain |
|---|---|---|---|
| RPC checkout | 30,000 | Image | |
| Twitter100k | 100,000 | | |
| INRIA-Websearch | 71,418 | Image-Text | Product |
| Dress Retrieval | 20,200 | | |
| Product1M | 1,182,083 | | |
| WIPO-alpha | 75,250 | | |
| CLEF-IP | 1,500,000 | | |
| USPTO-2M | 2,000,147 | Text | Patent |
| BIGPatent | 1,341,362 | | |
| HUPD | 4,518,263 | | |
| Support Product Set | 979,438 | Image-Text | Product |
| Gallery Patent Set | 13,410,443 | Image-Text | Patent |

and technical diagrams unrelated to infringement detection as shown in Figure 5 in Appendix C. These irrelevant images not only lack value, but also complicate the task, making the filtering and pre-processing of the image data crucial to improving the overall effective model training and detection accuracy. Here, we propose a simple yet effective model-based iteratively filtering method based on KNN (K-nearest neighbor), where our strategy achieves an overall recognition accuracy of 93% to successfully identify true noisy images, with a recall of 82.71% and a precision of 90.54%. Detailed pre-processing design can be found in Appendix C. In general, we provide statistics about the text and image sections as specified in Table 2, where more detailed description to each textual attribute can be found in Table 10 in Appendix. In addition, we highlight considerations, limitations, potential biases and ethic statement in Appendix E.

## 2.5 DATASET CHARACTERISTICS

**Significant Domain Gap.**   One of the most prominent challenges in our dataset is the significant domain gap between both vision and text representation. As shown in Figure 2, product images are typically captured as natural RGB photos, showcasing the items in real-world contexts. In contrast, patent illustrations often consist of black-and-white line drawings or schematics, which starkly differ in visual representation. This inherent discrepancy complicates cross-modal matching efforts. In terms of textual descriptions, product texts are generally brief and focused on essential attributes, while patent documents provide extensive structural information, including claims, backgrounds, and technical descriptions. This disparity in length and complexity makes direct similarity calculations between the two types of text particularly challenging, as shown in Table 2. Overall, these domain gaps necessitate robust methodologies to effectively bridge the differences, further enhance cross-domain representation learning and finally improve the accuracy of infringement detection.

**Large scale and Multi-purpose.**   In addition to tackling the product-patent infringement detection task, our dataset surpasses the size and diversity of previously available datasets in patent domains and achieve comparable size in product domains. As illustrated in Table 3, our collection marks a significant leap in scale. This expansive dataset not only facilitates comprehensive training for automated detection models but also supports a broader range of tasks, including multi-modal representation learning and language modeling (Radford et al., 2021; Devlin et al., 2019), patent classification (Larkey, 1999), product instance retrieval (Zhan et al., 2021) and etc. The substantial size of our dataset provides a robust foundation for developing models that generalize more effectively, addressing limitations in earlier datasets that were constrained by domain coverage.

**Consistency with Real-World Scenarios:**   The extensive 13-million patent retrieval pool in the Large version of our benchmark closely aligns with real-world conditions, presenting a formidable challenge for large-scale patent search and retrieval. This provides a realistic framework for evaluating the effectiveness of infringement detection models. Furthermore, both products and patents are accompanied by multiple images and drawings, along with complex textual structures as shown in Table 2. Leveraging all available visual and textual information for representing a product or patent can lead to substantial computational overhead. Therefore, selecting the most representative information is crucial for facilitating effective infringement detection. Additionally, we observe that number of samples in CPC main classes (gallery set of patent in ERiC-UP$^3$-Large dataset) follows a

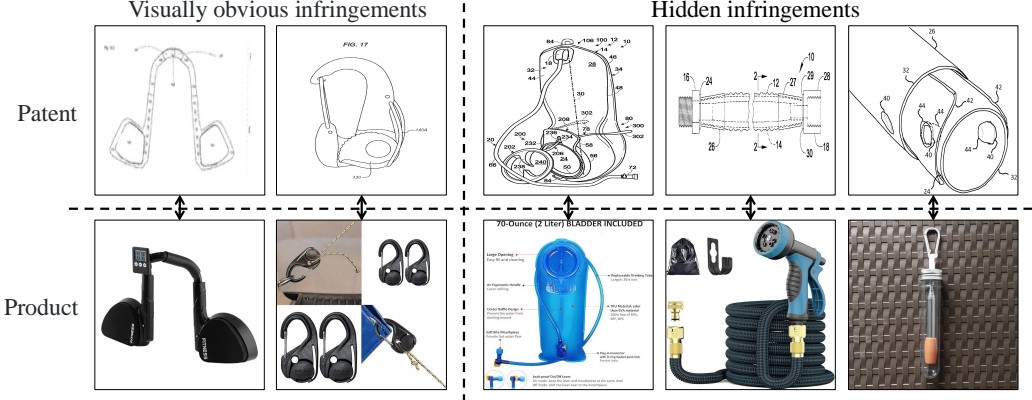

Figure 2: The significant domain gap between patent images (top row) and product images (bottom row) is evident, as patent images are typically black-and-white line drawings while product images are natural RGB photographs; some infringements are visually obvious, while more necessary the involvement of textual description for accurate detection.

heavily long-tail distribution as shown in Figure 6 in Appendix D, further indicating that our dataset effectively reflects real-world scenarios.

## 2.6 EVALUATION METRICS

To assess the performance of our infringement detection model, we focus primarily on two evaluation metrics: mean Average Top-$K$ matching Recall (mAR@K) and mean Rank of Matches (mRoM). Given that a single product may infringe on multiple patents, we adopt a hit-one strategy for recall evaluation. This means that if at least one infringing patent is present within the Top-$K$ matches, the detection is considered successful. Our objective is to achieve a lower average rank, indicating that relevant patents are retrieved more efficiently and effectively within the top results.

## 3 METHOD

In this section, we present a straightforward and effective pipeline for detecting potential patent infringements based on the **textual modality**. Given the enormous size of the patent retrieval pool (i.e., 13-million in ERiC-UP$^3$-Large), our primary concern is how to effectively reduce the search space. To address this, we design to employ a classifier to identify potential patent infringement categories related to products, thereby narrowing down the pool of relevant patents. Next, we design to train an encoder using supervised contrastive learning (SCL) (Khosla et al., 2020) on product-patent pairs, which generates reliable text embeddings, allowing us to compute similarity scores for infringement detection.

### 3.1 PATENT INFRINGEMENT CATEGORY CLASSIFIER

To handle the vast number of patents, our first step is to narrow down the search space, which can be achieved through a classifier that categorizes potential patent infringement types related to the given product. Formally, given a query product $P_i$, before retrieving potentially infringing patents, we first classify it into the relevant CPC main classes (137 in total) where potential infringing patents are likely to be found. This preliminary classification significantly reduces the search space, making the retrieval process more efficient.

Considering that a single product may potentially infringe upon multiple patents and that a patent can belong to multiple CPC main classes, we model this classification task as a multi-label classification problem. Specifically, for each product, we aim to predict the Top-$k$ CPC main classes that are most relevant. Let $\mathcal{C} = \{c_1, c_2, \ldots, c_N\}$ be the set of all CPC main classes, where $N$ is the total number of classes. Denote $T_i$ as the textual description of a product $P_i$ and $\mathbf{y}_i = [y_{i1}, y_{i2}, \ldots, y_{iN}]^\top$ as the ground truth label vector for product $P_i$, where $y_{ij} = 1$ if class $c_j$ is relevant to $P_i$, and $y_{ij} = 0$ otherwise. We use a neural network classifier $f_\theta$ parameterized by $\theta$ to predict the relevance scores for each class, $\hat{\mathbf{y}}_i = f_\theta(T_i) = [\hat{y}_{i1}, \hat{y}_{i2}, \ldots, \hat{y}_{iN}]^\top$, where $\hat{y}_{ij} \in [0, 1]$ represents the predicted probability that class $c_j$ is relevant to product $P_i$. $f_\theta$ can be optimized by minimizing a binary

cross-entropy (BCE) loss within a mini-batch $\mathcal{B}$ suitable for multi-label classification as below:

$$\arg\min_\theta -\frac{1}{|\mathcal{B}|} \sum_{i=1}^{|\mathcal{B}|} \sum_{j=1}^{N} \left[ y_{ij} \cdot \log(\hat{y}_{ij}) + (1 - y_{ij}) \cdot \log(1 - \hat{y}_{ij}) \right]. \tag{1}$$

During inference, for each product $P_i$, we compute the predicted probabilities $\hat{\mathbf{y}}_i = f_\theta(T_i)$, where $T_i$ is the textual description of $P_i$ and $f_\theta$ is the trained classifier. To determine the set of predicted CPC main classes, we employ both Top-$K$ selection and thresholding to ensure that only the most relevant and confidently predicted classes are considered.

Firstly, we select the Top-$K$ classes with the highest predicted probabilities, denoted as $C_i^{\text{Top-}K}$. Simultaneously, we apply a probability threshold $\lambda \in [0, 1]$ to include classes where the predicted probability meets or exceeds $\lambda$, forming the set $C_i^\lambda = \{c_j \in \mathcal{C} \mid \hat{y}_{ij} \geq \lambda\}$. The intersection of these two sets yields the final predicted classes: $C_i^{\text{Final}} = \begin{cases} C_i^{\text{Top-}K} \cap C_i^\lambda, & \text{if non-empty.} \\ C_i^{\text{Top-}K}, & \text{otherwise.} \end{cases}$ This combined approach leverages the consistency of Top-$K$ selection and the confidence provided by thresholding, enhancing both efficiency and accuracy. Using the final predicted classes $C_i^{\text{Final}}$, we reduce the patent retrieval pool for each product $P_i$ to patents classified under corresponding CPC main classes. By focusing on these subsets of the entire patent database, we significantly decrease computational requirements, improve the efficiency of subsequent retrieval steps and effectively narrows down the search space, facilitating the retrieval with higher mAR@K and lower mRoM.

## 3.2 Product-Patent Embedding Retriever

To effectively detect potential patent infringements based on textual content, we develop a **Product-Patent Embedding Retriever** that generates robust embeddings for both products and patents. Leveraging our training data consisting of product-patent pairs—where each pair includes a query product and a positive sample (an infringing patent)—we train our model to capture semantic similarities indicative of infringement relationships. By calculating the similarity between these embeddings, we can efficiently identify potential infringements.

Let $\mathcal{D} = \{(P_i, Q_i^+)\}$ denote the set of product-patent pairs in our training data, where $P_i$ is a product with textual description $T_{P_i}$ and $Q_i^+$ is the corresponding infringing patent (positive sample) with textual content $T_{Q_i^+}$. We employ a shared encoder $E_\phi$, parameterized by $\phi$, to map textual inputs into a latent embedding space $\mathbf{h}_{P_i} = E_\phi(T_{P_i}) \in \mathbb{R}^d$ and $\mathbf{h}_{Q_i^+} = E_\phi(T_{Q_i^+}) \in \mathbb{R}^d$, where $d$ is the dimensionality of the embedding space. Usually, $E_\phi$ can be initialized by well pre-trained language model such as BERT (Devlin et al., 2019) and RoBERTa (Liu, 2019).

Given that our training data consists of positive product-patent pairs, we aim to train the encoder such that embeddings of positive pairs are close in the latent space, while embeddings of negative pairs are pushed apart. Negative samples are crucial for effective training; we generate them by pairing each product with patents not associated with it. For each product $P_i$, we construct negative patents $Q_{i,j}^-$ by sampling from the reduced search space (obtained from the Patent Infringement Category Classifier), such that $Q_{i,j}^- \notin \{Q_i^+\}$. Finally, we employ a SCL approach using the InfoNCE loss (Oord et al., 2018) to effectively optimize the encoder $E_\phi$ as follows:

$$\arg\min_\phi -\frac{1}{|\mathcal{B}|} \sum_{i=1}^{|\mathcal{B}|} -\log \frac{\exp(\text{sim}(\mathbf{h}_{P_i}, \mathbf{h}_{Q_i^+})/\tau)}{\exp(\text{sim}(\mathbf{h}_{P_i}, \mathbf{h}_{Q_i^+})/\tau) + \sum_j \exp(\text{sim}(\mathbf{h}_{P_i}, \mathbf{h}_{Q_{i,j}^-})/\tau)}, \tag{2}$$

where $\text{sim}(\mathbf{h}_{P_i}, \mathbf{h}_Q) = \frac{\mathbf{h}_{P_i}^\top \mathbf{h}_Q}{\|\mathbf{h}_{P_i}\| \cdot \|\mathbf{h}_Q\|}$ is the cosine similarity between embeddings and $\tau$ is a temperature hyper-parameter controlling the concentration level of the distribution. Furthermore, inspired by hard-sample mining (Karpukhin et al., 2020) we design to periodically update the negative samples with patents that the model currently finds challenging, enhancing discrimination.

During inference, we firstly generate all candidate patent embeddings $\{\mathbf{h}_{Q_j}\}$ for all patents $Q_j$ in the reduced search spac, where the pre-computing and indexing these embeddings accelerates retrieval. Then for each product $P_i$, we can compute the product embedding $\mathbf{h}_{P_i} = E_\phi(T_{P_i})$ and calculate the cosine similarity between $\mathbf{h}_{P_i}$ and each $\mathbf{h}_{Q_j}$ by $s_{i,j} = \text{sim}(\mathbf{h}_{P_i}, \mathbf{h}_{Q_j})$. As a result, we can

Table 4: Retrieval performance for different combinations of product and patent textual sections on the validation set of the A63 CPC main classes. We mainly focus on higher mAR@500 and then lower mRoM. More comprehensive results can be found in Appendix F.1.

| Product \ Patent | Tit. mAR@500/ mRoM | Abs. mAR@500 / mRoM | Bkg. mAR@500 / mRoM | CL. mAR@500 / mRoM | Abs. + CL. mAR@500 / mRoM | Tit. + Abs. + CL. + Bkg. mAR@500 / mRoM |
|---|---|---|---|---|---|---|
| Tit. | 23.21 / 116.57 | 57.14 / 238.33 | 19.64 / 383.01 | 30.36 / 215.07 | 48.21 / 283.01 | 33.93 / 146.17 |
| Desc. | 30.36 / 156.67 | 57.14 / 188.76 | 21.43 / 279.88 | 41.07 / 280.21 | 58.93 / 254.79 | 57.14 / 265.14 |
| Tit. + Desc. | 42.86 / 271.44 | 53.57 / 208.28 | 26.79 / 220.38 | 50.00 / 272.82 | 60.71 / 235.34 | 60.71 / 238.11 |

finally obtain a ranked list $R_i$ based on similarity scores $s_{i,j}$ in descending order and evaluate the effectiveness of the method by checking whether ground truth patent in Top-$K$ of $R_i$.

By fine-tuning an encoder using supervised contrastive learning on our annotated product-patent pairs, we effectively capture the semantic relationships necessary for infringement detection. The embeddings generated by our Product-Patent Embedding Retriever enable efficient and accurate identification of potential infringements based on textual similarity.

## 4 EXPERIMENTS AND ANALYSIS

In this section, we conduct comprehensive experiments on our datasets. Due to the substantial training costs and computational demands, we focus on conducting a series of baseline experiments on ERiC-UP$^3$-Base, and ultimately provide results on ERiC-UP$^3$-Large. Unless otherwise specified, our experimental results are primarily based on the text modality, where we consider mean Average Top-500 matching Recall (mAR@500) and mean Rank of the Matches (mRoM). We highlight the best performance in *red* and the second one in *blue*.

### 4.1 WARMINGUP: SELECTION OF TEXTUAL SECTIONS AND ENCODER

To establish a strong baseline for our retrieval model, we begin by exploring the optimal combination of textual sections from patents and products, as well as selecting the most suitable encoder for our task. This preliminary step is crucial to ensure that subsequent experiments would be built upon the most informative and effective data representations. Patents and products contain various textual components that provide different levels of detail and specificity.

As shown in Table 2, both patent and product include several textual sections. We hypothesize that certain combinations of these sections might offer better retrieval performance due to the richness of information they contain. Therefore, we examine performance of different combinations of these sections as shown in Table 4, which experimentally demonstrate that the combination of *Abstract + Claims* for patents and *Title + Description* for products is optimal by achieving a highest mAR@500 score of 60.71 and an mRoM of 235.34. This combination can effectively capture essential legal and technical aspects, and finally enhance the retrieval system's effectiveness. We put implementation details and analysis in Appendix F.1.

By following common practice of text multi-class classification, we employ a pre-trained language model and a linear layer to serve as $f_\theta$. For initialization of encoder used in classifier $f_\theta$ and retriever $E_\phi$, we examine several popular pre-trained models, including BERT (Devlin et al., 2019), RoBERTa (Liu, 2019), T5 (Raffel et al., 2020), MPNET (Song et al., 2020), BGE (Xiao et al., 2023) and LlaMa2-7B (Dubey et al., 2024). As shown in the top cell of Table 5, we find that both RoBERTa-large and BGE-large achieved comparable and satisfactory performance with similar parameter sizes and structures (around 350M). It is not surprising that T5-large achieves a more significant mAR@500, due to about 770M parameters and an encoder-decoder structure. Here, *ALL* indicates that we use all five categories of patents as the retrieval pool to evaluate the performance of matching. The remaining results represent the performance under the assumption that we already know the main CPC classes of the patent infringed by the current product. Ultimately, considering the model structure, efficiency and performance, we employ BGE to initialize encoders in $f_\theta$ and $E_\phi$. More details and discussion can be found in Appendix F.1.

### 4.2 HOT TO EFFECTIVELY REDUCE PATENT SEARCH SPACE?

One of the biggest challenges in training a robust Infringement Category Classifier is the lack of a large-scale training set with labels from product to patent CPC main classes, which may result in the classifier overfit and lack of enough generalization ability due to the scarcity of training data.

Table 5: mAR@500 of different pre-trained (top) and fine-tuned encoders (bottom).

| Encoder | ERiC-UP$^3$-Base | | | | | |
|---|---|---|---|---|---|---|
| | ALL | A45 | A47 | A63 | B65 | H01 |
| BERT-large | 8.81 | 14.29 | 26.92 | 14.49 | 17.07 | 11.57 |
| RoBERTa-large | 16.08 | 22.69 | 18.12 | 17.14 | 12.20 | 23.14 |
| T5-large | 22.47 | 31.73 | 31.16 | 25.71 | 17.07 | 28.93 |
| MPNET-large | 13.42 | 12.50 | 26.09 | 15.71 | 13.41 | 25.62 |
| BGE-large | 16.08 | 14.42 | 26.81 | 20.00 | 18.54 | 29.01 |
| LLaMa2-7B | 5.73 | 2.88 | 7.23 | 5.71 | 13.41 | 9.09 |
| OURS (RoBERTa) | 45.81 | 66.35 | 63.04 | 40.00 | 50.00 | 40.50 |
| OURS (T5) | 64.54 | 92.31 | 68.12 | 60.00 | 59.76 | 56.20 |
| OURS (BGE) | 68.32 | 75.00 | 73.19 | 62.86 | 75.61 | 59.60 |

Table 6: Performance of different training datasets on CPC classification

| Test Dataset Partial Accuracy (%) | ERiC-UP$^3$-Base | | |
|---|---|---|---|
| | Top-1 | Top-2 | - |
| Infringement Pairs | 85.01 | 88.87 | - |
| GPT-4 Generation | 75.22 | 82.01 | - |
| Patent Classification | 90.01 | 95.81 | - |

| Test Dataset Partial Accuracy (%) | ERiC-UP$^3$-Large | | |
|---|---|---|---|
| | Top-1 | Top-2 | Top-5 |
| Infringement Pairs | 33.79 | 50.06 | 61.28 |
| GPT-4 Generation | 57.67 | 70.63 | 78.12 |
| Patent Classification | 73.87 | 85.60 | 91.01 |

Therefore, we propose two new methods to construct the labels for the product to patent CPC main classes, *1)* based on GPT-4 to generate the training set; *2)* based on patent CPC classification. We set Top-$K = 5$ and $\lambda = 0.2$ during the evaluation, and provide construction details, ablation study and discussions in Appendix F.2. As illustrated in Table 6, we primarily consider Top-1&2 accuracy on the *Base* test set (including 5 classes) , while on the *Large* test set (137 classes), we consider Top-1,2&5. It is evident that classifiers trained on infringement pairs often do not perform optimally, and classifiers trained on the GPT-4 generated training set also fail to effectively map products to the infringement patent CPC main classes. Interestingly, we find that classifiers trained on patent data (mapping patent text to corresponding CPC main classes) exhibit excellent transferability and generalization capabilities, which can effectively categorize products into main classes of potentially infringing patents, demonstrating a highly effective and stable method of reducing the patent search space. This discovery underscores the importance of leveraging patent data in training robust classifiers, where the inherent structure and rich information contained in the patent provide a solid foundation for the classifier to learn meaningful mappings from products to patent classes.

## 4.3 TRAINING AN EFFECTIVE PRODUCT-PATENT EMBEDDING RETRIEVER

To evaluate the effectiveness of our proposed product-patent embedding retriever, we conduct a series of experiments. The objective of these experiments is to understand how well our model can learn meaningful embeddings that capture the relationship between products and patents and how these embeddings can be used to identify potential patent infringements. The results in bottom block of Table 5 show that with the help of supervised contrastive learning by minimizing Equation 3.2, our fine tuned model is able to effectively learn meaningful embeddings that capture the relationship between products and patents, outperforming several baseline methods by 29.73% on RoBERTa, 42.07% on T5 and 52.24% on BGE. Furthermore, the embeddings can be used to efficiently retrieve relevant patent documents for a given product description, demonstrating the potential of our model for automated patent infringement detection. These experiments demonstrate the effectiveness of our proposed product-patent embedding retriever. Our model not only provides a way to automatically detect potential patent infringements but also opens up new possibilities for further research in the field of patent analysis and classification. Finally, we report the performance of our pipeline in Table 7, from which we can observe that with the help of Classifier and Retriever, our pipeline can effectively improve mAR@500 by 24.05% and 28.37%, where mRoM also significantly decreases from 110.00 to 102.28 and 124.75 to 93.15, respectively.

## 4.4 ANALYSIS

**Rewritten for Better Textual Matching.** In the context of product and patent retrieval, effective text matching is crucial for identifying semantically similar content across different documents. However, challenges such as varying text lengths, redundant information, and significant stylistic differences between patent and product descriptions can hinder the performance of embedding models. To address these issues, we employ two rewriting strategies: (1) summarizing long texts into shorter, more focused contexts, and (2) aligning the stylistic differences between patent and product texts. We utilize several open-source Large Language Models (LLMs) for these rewriting tasks. Experimental results for mAR@500 are presented in Table 8, with additional results and analyses provided in Table 14 in Appendix F.3. Through these experiments, we find that rewriting is an effective strategy for improving text matching, with summarization proving to be more robust than stylistic alignment. These findings indicate that further exploration of these techniques could lead to even better outcomes, offering an alternative approach for researchers to consider.

Table 7: Final results on both Large and Base test set.

| | Pool | mAR@500 | mRoM |
|---|---|---|---|
| Base Test | 1300W | 39.38 | 110.00 |
| | OURS | 63.43 | 102.28 |
| Large Test | 1300W | 26.22 | 124.75 |
| | OURS | 54.59 | 93.15 |

Table 8: mAR@500 results of various LLMs on the three rewritten subsets based on pre-trained BGE-large.

| Subset | Base | Summary | | | Stylistic-Align. |
| | | Qwen2-0.5B | Qwen2-7B | Llama3-8B | Qwen2-0.5B |
|---|---|---|---|---|---|
| A45 | 14.42 | 27.88 | 14.42 | 31.73 | 8.65 |
| A47 | 26.81 | 32.61 | 31.88 | 34.06 | 27.54 |
| A63 | 20.00 | 32.86 | 35.71 | 31.43 | 34.29 |

**Patent** ... substantial domain disparities → *Stretch Detection* → similar visual styles ... **Product**

Figure 3: Visualizations of stretch extraction for both patent and product images. This strategy significantly mitigates domain shift, leading to improved image retrieval performance.

**Image-Retrieval Based on Stretch.** As shown in Figure 3, we propose a simple yet effective style-transfer method based on stretch detection (Zhou et al., 2024) to alleviate the domain shift between patent and product images, and then utilize the powerful CLIP (Radford et al., 2021) model to extract unified feature representations for similarity measurement. Results in Table 9 demonstrate that, benefiting from stretch-based style

Table 9: mAR@500 results of image-retrieval and cross-modal retrieval.

| Method | mAR@500↑ |
|---|---|
| Raw natural-style | 12.50 |
| Stretch-based-style (OURS) | 33.92 |
| Text-to-Image (OURS) | 42.85 |
| Cross-Modality (OURS) | 57.14 |

transfer, we achieve a mAR@500 of 33.92%, marking a 21.42% improvement over using raw natural images. Considering the remarkable performance and low computational cost of text-based retrieval, we further propose a *hierarchical text-to-image retrieval strategy*. In the first stage, we utilize text matching to filter the Top-$h$ most likely patent candidates for each product. Next, we perform image-based retrieval within the selected Top-$h$ candidates. With this approach, we further enhance image-based retrieval performance, increasing mAR@500 from 33.92% to 42.85%. Details can be found in Appendix F.4.

**Cross-Retrieval Based on CLIP.** Following the hierarchical text-to-image retrieval framework, we first employ text matching to select the Top-$h$ most likely patent candidates for each product. Subsequently, rather than using stretch-based image retrieval, we further explore the effectiveness of cross-modal retrieval using the CLIP model by computing the similarity between the encoded text features of product and the stretch image features of patent candidates. As showcased in Table 9, the cross-modal text-to-image retrieval performance, enabled by CLIP's robust text-image alignment capabilities, achieves a mAR@500 of 57.14%, surpassing the sketch-based image retrieval by a notable 14.29% margin. This result indicates that adaptively selecting the optimal retrieval modality and multi-modal fusion mechanism for each sample may be critical to achieving effective patent-product infringement detection.

**Does Cosine Similarity Best Capture Semantic Relevance?** Several works (Khattab & Zaharia, 2020; Lu et al., 2021; Steck et al., 2024) have highlighted that simple dot-product (or cosine similarity) between embeddings may not be sufficient to capture semantic relevance. To address this, we propose an alternative metric to replace naive cosine similarity as the detector. Inspired by supervised fine-tuning (SFT), we use paired product-patent embeddings with binary labels to train a two-category classifier. During inference, the output logits are used as the metric to measure the semantic relevance between the paired product and patent embeddings. The experimental results and detailed analyses presented in Appendix F.5 also suggest that cosine similarity may not be the most effective detector for patent infringement.

**Visual-Enhanced Multi-Modality Infringement Detection.** In this section, we present an effective analysis on how to fuse textual features and visual features to achieve efficient infringement detection. The details and experimental analysis are provided in the Appendix F.6. Our final con-

clusion is that visual and textual information complement each other in the detection, leading to better results. However, naive fusion cannot achieve the best performance, necessitating the design of superior fusion or voting methods to balance the importance of visual and textual features.

# 5 RELATED WORK

**Information Retrieval.** Information retrieval, encompassing both intra-modal and cross-modal techniques, plays a pivotal role in efficiently accessing relevant data from vast information sources. Intra-modal retrieval (Qi et al., 2016; Bai et al., 2018) has been thoroughly explored in various domains, such as keyword-based web document search (Ensan & Bagheri, 2017), content-driven image retrieval (Noh et al., 2017) and product recommendation systems (Kang et al., 2017). On the other hand, cross-modal retrieval (Feng et al., 2014; Wang et al., 2017) has emerged as a compelling solution for efficiently indexing and retrieving data across different modalities, making it particularly useful in large-scale applications like search engines (Harman et al., 2019) and E-commerce platforms (Corbiere et al., 2017), among others. Nevertheless, these techniques (Nurmi et al., 2008; Wang et al., 2016; Lin et al., 2018) often rely on single-modal inputs, limiting their effectiveness in real-world scenarios where both queries and targets involve multi-modal information.

**Patent Analysis.** Previous research in Natural Language Processing (NLP) related to patent analysis has largely concentrated on two key tasks: patent classification and summarization. Patents are typically classified using hierarchical systems such as the IPC and CPC, with various studies predicting IPC/CPC codes at different levels using statistical methods (Chu et al., 2008; Tran & Kavuluru, 2017; Gomez, 2019) and neural networks (Grawe et al., 2017; Li et al., 2018; Zhu et al., 2020), including Transformer-based models like BERT (Devlin et al., 2019) and BIGBIRD (Zaheer et al., 2020). Datasets such as CLEF-IP (Piroi et al., 2011) and USPTO-2M (Li et al., 2018) have been commonly used for training models in this area, though they are limited in scope and flexibility, which the more comprehensive HUPD (Suzgun et al., 2024) dataset addresses. In the area of patent text generation and summarization, the introduction of the BIGPATENT (Sharma et al., 2019) dataset marked a significant step forward. This work introduces a novel task by focusing on patent acceptance prediction, using textual analysis to identify characteristics that differentiate accepted from rejected patents, thus contributing a new dimension to patent decision classification.

**Patent Infringement Detection.** Patent infringement detection is a crucial process aimed at identifying the unauthorized use of patented technology, thus safeguarding IP rights. Traditionally, this task has been carried out manually, requiring detailed comparisons of patent claims—a method that is both time-consuming and susceptible to human error (Schoen et al., 1993; Majewski & Williamson, 2004). In recent years, keyword-driven text mining techniques have become a prevalent approach for detecting infringements (Yoon, 2008; Lee et al., 2013). However, these techniques are limited by their dependence on predetermined keywords, which constrains their ability to capture nuanced technological insights and complex structural relationships between components. To address these limitations, Park & Yoon (2014) proposed a semantic similarity approach based on the Subject-Action-Object (SAO) framework, utilizing WordNet (Miller, 1995) to identify patent infringements by measuring technological similarities. More recently, with the advancement of deep neural networks, Liu & Pei (2023) leverages text vectorization and convolutional neural networks to extract and represent patent infringement features, capturing semantic information from multiple layers of patents.

# 6 CONCLUSION

In this paper, we introduce a new task of detecting potentially infringing patents for given products represented by multi-modal data, including both images and textual descriptions. To support this, we develop ERiC-UP[3], the largest and most comprehensive dataset for this task. Our experiments highlight the complexity of the problem and demonstrate the potential of our detection pipeline, along with techniques like text style rewriting and cross-modal matching, to improve results. This work establishes a foundation for advancing automated IP infringement detection, helping mitigate legal risks and foster innovation across industries.

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

# Appendix

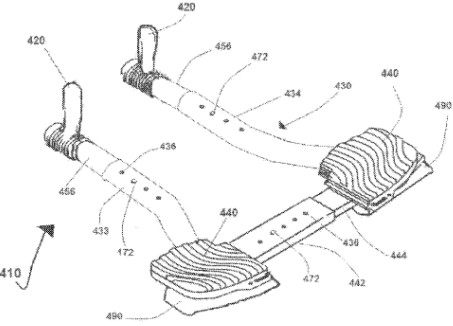

Figure 4: Example of a patent (No.US10286245B2, page one).

## A ZOOMED-IN EXAMPLE

## B DIFFERENCE WITH PREVIOUS RELATED DATASETS

The ERiC-UP[3] dataset showcases significant advantages from the perspectives of patents, products, and annoatated infringement pairs. Firstly, the patent data is more comprehensive and larger in scale, encompassing a vast amount of technical texts and diagrams, which provide rich technical details for models. Secondly, the product data is more diverse and abundant, containing a large number

of product images and descriptions. This not only allows for the completion of tasks supported by previous datasets but also enables more complex applications. Thirdly, the infringement pairs are meticulously annotated and have undergone multiple rounds of expert validation, ensuring the accuracy and reliability of the data.

Most importantly, ERiC-UP[3] focuses on the new task of **infringement detection**, filling a gap present in existing datasets. By providing more comprehensive, larger-scale, and high-quality patent-product paired data, we offer robust support for researchers and practitioners working at the intersection of e-commerce and IP law, thereby advancing research and application development in this field.

## B.1 COMPARISON WITH PATENT-RELATED DATASETS

When comparing our dataset to existing patent datasets such as HUPD (Suzgun et al., 2024) and BIGPATENT (Sharma et al., 2019), several key distinctions highlight the unique contributions of our work.

**Scope and Purpose:** Our dataset is specifically designed for the task of patent infringement detection, providing data tailored to this complex and nuanced challenge. In contrast, BIGPATENT is primarily aimed at abstractive summarization tasks and includes only a limited selection of data fields—specifically, publication number, application number, abstract, and description. It notably lacks the **claims** section, which is critical for understanding the specific legal and technical assertions of a patent. HUPD offers more fields than BIGPATENT but is still not tailored for infringement detection.

**Data Richness and Comprehensiveness:** Our dataset includes a much richer set of bibliographic metadata and full patent documents, encompassing all essential sections such as abstracts, descriptions, claims, summary, CPC code and publication month. This comprehensive inclusion supports a wide array of analyses and facilitates more in-depth research into patent infringement.

**Introduction of Multi-modal Data—A Major Breakthrough:** A significant advancement of our dataset is the incorporation of multi-modal data. Unlike HUPD and previous datasets that are solely text-based, we have collected and cleaned the figures and drawings from patents. This is a substantial contribution, as it enables models to learn from both textual and visual information, providing a more holistic understanding of patents. The inclusion of images opens up new research possibilities in multi-modal machine learning applications within the patent domain.

**Largest Scale and Broad Applicability:** Our dataset is the largest available, surpassing previous datasets in both size and depth of information, where we release 13 million patent data and that is 4.5 million in HUPD and 1.3 million in BIGPATENT. This extensive scale supports the development of more robust and generalizable machine learning models.

**Data Processing Flexibility:** Some existing datasets provide text that has been pre-tokenized or processed in ways that may inadvertently introduce issues, especially with complex content like chemical formulas or mathematical equations—BIGPATENT, for example, is pre-tokenized using NLTK. Our dataset, however, provides the **raw patent text**, allowing researchers to apply custom tokenization and preprocessing techniques suitable for accurately handling specialized technical content.

**Task Enablement:** The richness and structure of our dataset enable new research directions and tasks that were previously challenging or unattainable. This includes fine-grained classification, temporal analysis of patent texts, and more sophisticated infringement detection methods that leverage the full depth of information contained within patents.

Our dataset overcomes the limitations of existing resources like HUPD by providing a more comprehensive, multi-modal resource that includes crucial sections like the claims and incorporates cleaned figures and drawints from patents. It stands out as the largest and most versatile dataset, supporting all existing tasks and introducing new ones, such as patent infringement detection. This combination of scale, depth, and the groundbreaking inclusion of visual data represents a significant leap forward in patent analysis and machine learning applications in this field.

Table 10: Descriptions and statistics of patent and product textual sections, and average number of images / drawings per sample.

| | Section | Brief Description | ERiC-UP[3]-Base Avg # Words | ERiC-UP[3]-Large Avg # Words |
|---|---|---|---|---|
| Gallery set of Patent | Title | Succinctly describes the invention. | 7.6 | 7.77 |
| | Abstract | Provides a brief summary of the invention's key points. | 111.01 | 104.28 |
| | Claims | Define the scope of the patent protection. | 894.48 | 946.61 |
| | Background | Explains the context and prior art related to the invention. | 99.65 | 96.83 |
| | CPC Code | CPC code categorizes the patent. | - | - |
| | Publication Number | A unique identifier assigned to the published patent application. | - | - |
| | Publication Month | Indicates when the patent application was published. | - | - |
| | #imgs / sample | Average number of images per patent. | 21.01 | 20.51 |
| Support set of Product | Title | Provides a concise name for the item. | 11.72 | |
| | Description | Detailed information about the product's features and benefits. | 122.83 | |
| | #drawings / sample | Average number of drawings per product. | 10.77 | |

### B.2 COMPARISON WITH PRODUCT-RELATED DATASETS

When comparing our dataset to existing product datasets such as RPC (Wei et al., 2019), Twitter100k (Hu et al., 2017), INRIA-Websearch (Krapac et al., 2010), Dress Retrieval (Corbiere et al., 2017), and Product1M (Zhan et al., 2021), several key distinctions highlight the unique contributions of our work:

**Focus on Product-Patent Infringement Detection:** ERiC-UP[3] is the first dataset specifically designed for product-patent infringement detection, addressing a critical need at the intersection of e-commerce and IP law. Other datasets focus on related but fundamentally different tasks: RPC focuses on product recognition in retail checkout scenarios without involving patent data or infringement detection. Twitter100k contains informal image-text pairs for cross-media retrieval but does not involve products in a legal context or any patent information. INRIA-Websearch deals with general cross-modal retrieval related to broad queries like actors and films, lacking a focus on products or patents. Dress Retrieval targets fashion image retrieval with associated textual attributes but does not address patent data or infringement issues. Product1M is designed for instance-level, multimodal product retrieval in e-commerce but does not encompass patent information or infringement detection.

**Scale and Diversity with Practical Relevance:** Our dataset offers an additional 1 million multimodal product samples to support effective product representation learning. This scale and specificity is one of the largest product-related datasets.

## C NOISY IMAGES CONTAINED IN BOTH PATENT AND PRODUCT SAMPLES

This section highlights the importance of differentiating between essential drawings and images for infringement detection, represented by the "Greed Box" and less relevant technical illustrations or background images in the "Red Box". The former showcases key design features, while the latter includes structural diagrams that do not contribute to infringement analysis and should be excluded from consideration.

To this end, we propose a simple yet effective model-based iteratively filtering method based on KNN (K-nearest neighbor), where we firstly employ DINO (Oquab et al., 2023), a widely recognized unsupervised pre-training method, to train a robust feature extractor on large-scale patent/product images. Next, we curate a small hand-labeled dataset, denoted as $\phi$, with labels indicating whether images are noisy or valid, and generate the feature embedding for the set $\phi$ using DINO. Upon completing these preparations, for each image to be classified, we compute its feature embeddings and retrieve the Top-$K$ most similar instances from $\phi$. By voting on the labels of the Top-$K$ instances, we predict whether an image is noisy. The predicted noisy and valid images are then added to $\phi$, iteratively expanding the labeled dataset and enabling more robust noise prediction. To validate the effectiveness of our approach, we construct a test set containing 81 noisy images and 219 valid images. Our KNN strategy achieves an overall recognition accuracy of 93%, successfully identifying 67 true noisy images, with a recall of 82.71% and a precision of 90.54%.

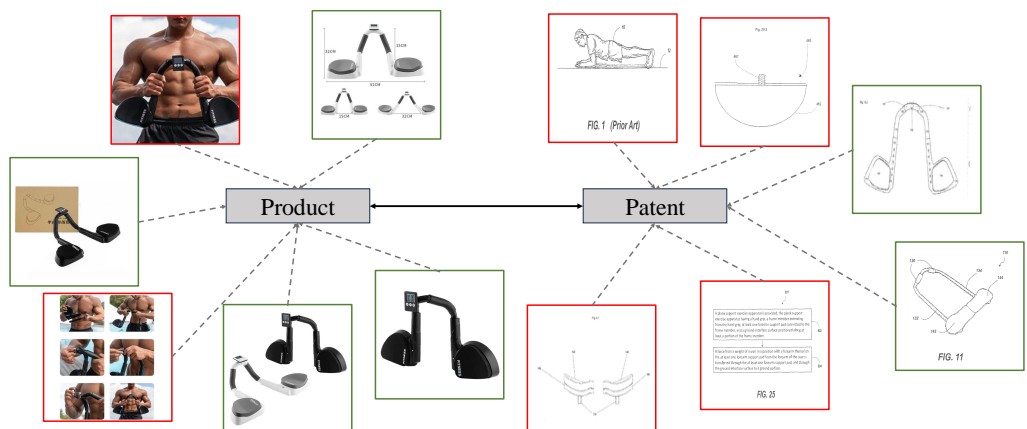

Figure 5: *Greed Box:* drawings that significantly help infringement detection, showing the core design and functional features. *Red Box:* drawings of partial structures or technical flow diagrams, which, despite being prominent in the patent images, are not valuable for infringement detection and should be filtered out.

## D   ANALYSIS OF SAMPLE DISTRIBUTION IN CPC CLASSES

Table 11: Number of samples in the gallery set of patent in ERiC-UP$^3$-Large and ERiC-UP$^3$-Base.

| Version | ERiC-UP$^3$-Large | | | | | | | | |
|---|---|---|---|---|---|---|---|---|---|
| | A | B | C | D | E | F | G | H | Y |
| Number of Samples | 3,476,219 | 3,225,770 | 2,636,832 | 169,288 | 453,190 | 1,571,989 | 6,456,432 | 6,149,238 | 1,025,546 |

| Version | ERiC-UP$^3$-Base | | | | | | |
|---|---|---|---|---|---|---|---|
| | A45 | A47 | A63 | B65 | H01 | | - |
| Number of Samples | 63,681 | 217,970 | 190,675 | 307,749 | 1,771,767 | | - |

This section presents a comprehensive analysis of the distribution of samples across various CPC main classes. The bar chart (a) illustrates the number of samples sorted by ID, highlighting the concentration of samples in specific classes. The pie charts (b) and (c) depict the proportions of different CPC sections for the ERiC-UP$^3$-Large and ERiC-UP$^3$-Base datasets, respectively, showcasing the relative significance of each section in the overall dataset. Specific number of samples of Figure 6 (b) and Figure 6 (c) are shown in Table 11.

## E   CONSIDERATIONS AND ETHICAL INSIGHTS

The dataset was created to build new and useful benchmarks for IP experiments, facilitate research on IP protection, patent-product infringement, patent and product analysis, and eventually help small entities and businesses proactively detect and mitigate potential infringements to minimizing risks, ensuring smoother operations and fostering continued innovation. We highlight our limitations, potential biases, ethical statements and distribution of the dataset as below.

**Limitations:** Methodologically, our dataset is confined to English and omits certain textual sections, such as the inventor information in patents and product categories. Additionally, some textual elements exceed the processing limits of current NLP models, and specialized vocabulary in specific fields can pose challenges for existing tokenizers. In addition, as shown in Figure 5, some noisy images are not avoidable in the process of data collection.

**Potential Biases:** The labeling process for training and test pairs relies heavily on the expertise of patent specialists, which introduces the possibility of erroneous annotations. Although we implemented a multi-expert review process with multiple rounds of discussion to ensure data quality, some inaccuracies may still persist.

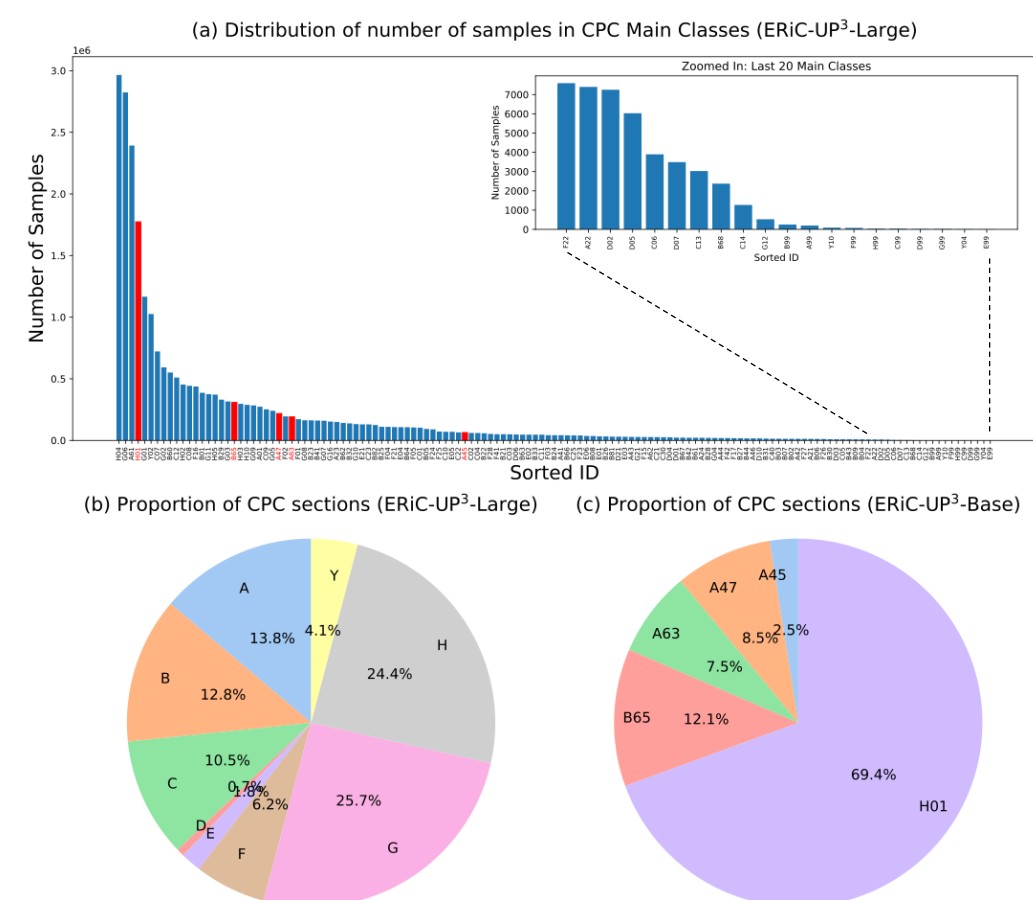

Figure 6: Illustration of the distribution of samples within CPC main classes and the proportions of various CPC sections for two datasets, ERiC-UP$^3$-Large and ERiC-UP$^3$-Base. The bar chart (a) shows the number of samples per class, while the pie charts (b) and (c) provide a breakdown of the proportions of different CPC sections, highlighting the varying representation within the datasets.

**Ethical Statements:** In constructing the ERiC-UP$^3$ benchmark and dataset, we follow the data statement guidelines outlined by Gebru et al. (2021); Bender & Friedman (2018), which includes discussing our motivations, objectives, collection processes, workflows, use cases, distribution, potential contributions, and any associated challenges.

**Distribution and Maintenance:** We will publicly release all the data, along with code and fine-tuned models upon acceptance.

## F   DETAILED EXPERIMENTS AND ANALYSIS

We conduct all the experiments on 2*8 Tesla V100 GPU cards except for rewritten task, which are conducted on 2*8 Tesla A100 GPU cards.

### F.1   WARMINGUP: SELECTION OF TEXTUAL SECTIONS

Based on the A63 subclass that includes 56 infringement test pairs as a validation, we evaluate the retrieval performance of different section combinations measured by mAR@500 and mRoM as shown in Table 4. The results demonstrate that combining the *Abstract* and *Claims* sections of patents with the *Title* and *Description* of products yield the best results, achieving a highest mAR@500 score of

60.71 and an mRoM of 235.34. While including all patent sections (Title, Abstract, Background, and Claims) also reachs the same mAR@500 score and similar mRoM, it results in significantly longer inputs that often exceeded the encoder's processing capacity and introduced additional noise. Therefore, considering both performance and computational efficiency, we conclude that the combination of *Abstract + Claims* for patents and *Title + Description* for products is optimal, effectively capturing essential legal and technical aspects and enhancing the retrieval system's effectiveness. This conclusion is also supported by the results on A47 CPC main classes as shown in Table 12.

Table 12: Retrieval performance for different combinations of product and patent textual sections on the test set of A63 (top) and A47 (bottom) CPC main classes. We mainly focus on higher mAR@500 and then lower mRoM.

| Patent / Product | Tit. mAR@500 / mRoM | Abs. mAR@500 / mRoM | CL. mAR@500 / mRoM | Abs. + CL. mAR@500 / mRoM | Tit. + Abs. + CL. + Bkg. mAR@500 / mRoM |
|---|---|---|---|---|---|
| Title | 11.43 / 160.38 | 8.57 / 215.33 | 28.57 / 163.8 | 22.86 / 247.63 | 24.29 / 217.94 |
| Description | 22.86 / 186.13 | 4.29 / 149.67 | 32.86 / 196.22 | 22.86 / 187.63 | 35.71 / 232.72 |
| Title + Description | 28.57 / 228.15 | 8.57 / 272.67 | 24.29 / 186.65 | 37.14 / 236.46 | 31.43 / 203.96 |
| Title | 29.71 / 222.88 | 13.77 / 238.00 | 26.09 / 218.22 | 26.09 / 297.11 | 21.74 / 196.87 |
| Description | 15.94 / 202.05 | 23.19 / 184.78 | 23.19 / 228.57 | 23.19 / 228.56 | 23.91 / 248.85 |
| Title + Description | 27.54 / 206.42 | 15.22 / 183.43 | 26.09 / 196.28 | 31.88 / 168.27 | 24.64 / 177.38 |

### F.2 DETAILS OF TRAINING A ROBUST INFRINGEMENT CATEGORY CLASSIFIER

In this context, *all* indicates that we use all five categories of patents as the retrieval pool to evaluate the performance of our classifier. The remaining results represent the performance under the assumption that we already know the main CPC classes of the patent infringed by the current product. This distinction is crucial as it simulates two different real-world scenarios. The first scenario (*all*) is more challenging as it requires the classifier to correctly identify the relevant patent category among all available categories. The second scenario is less challenging as it assumes prior knowledge of the correct patent category, thus narrowing down the search space and potentially improving the classifier's performance.

Here, we firstly provide ablation study on using patent classification as patent infringement category classifier. The Table 13 presents the performance of *PATENT INFRINGEMENT CATEGORY CLASSIFIER* trained with varying ratios of 13 million patents. The metrics used to evaluate the performance include Partial Intersection Accuracy, Recall, Precision, and F1 score. We set Top-$K = 5$ and $\lambda = 0.2$.

Table 13: Performance of the Cooperative Patent Classification (CPC) classifier at different training data ratios. The table shows the Partial Intersection Accuracy, Recall, Precision, and F1 score for the classifier trained with different ratio of 13 million patents. The results highlight the positive correlation between the amount of training data and the performance of the classifier, with a particular emphasis on the Partial Intersection Accuracy metric.

| Ratio | Partial Intersection Accuracy | Recall | Precision | F1 |
|---|---|---|---|---|
| 1% | 59.90 | 26.89 | 27.22 | 27.01 |
| 5% | 72.78 | 28.80 | 30.79 | 27.87 |
| 10% | 85.60 | 57.87 | 50.21 | 53.77 |
| 30% | 91.01 | 56.16 | 62.32 | 59.08 |

- **Positive Correlation between Ratio and Performance:** The table clearly demonstrates a positive correlation between the ratio of data used for training and the performance of the classifier. As the ratio increases, all performance metrics improve. This suggests that the classifier benefits from more training data, which allows it to learn more complex representations and make more accurate predictions.
- **Focus on Partial Intersection Accuracy:** The primary metric of interest in this experiment is Partial Intersection Accuracy. This metric measures the accuracy of the classifier when

the prediction and the ground truth label share at least one common element (i.e., have a non-empty intersection). This is a more lenient measure of accuracy, as it allows for partial matches between the predicted and actual labels. It is particularly useful in multi-label classification tasks, where each instance can belong to multiple classes, and a prediction is considered correct as long as it identifies at least one correct class.

The results show that even with only 1% of the data, the classifier can achieve a Partial Intersection Accuracy of nearly 60%. This accuracy improves to over 91% when 30% of the data is used. These findings highlight the importance of having a large amount of training data for improving the performance of the CPC classifier, especially when evaluated using Partial Intersection Accuracy. They also underscore the utility of the Partial Intersection Accuracy metric for evaluating multi-label classification tasks.

Further, we investigate the impact of different Top-$K$ and $\lambda$ values on the final classification performance. The Top-$K$ parameter refers to the number of most likely classes that the classifier outputs. It is a key parameter in multi-label classification tasks, as it determines the granularity of the predictions. A larger K means that the classifier will predict more classes for each instance, potentially increasing recall but possibly decreasing precision if many of the extra predicted classes are incorrect as shown in Figure 7.

$\lambda$ is a threshold parameter that determines the number of activations required for a class to be considered as a potential prediction. A higher $\lambda$ means that more activations are needed for a class to be considered, which can increase precision (since only the most activated classes are considered), but may decrease recall (since some less activated but still relevant classes might be missed), as shown in Figure 8. As for the Top-$K$ parameter, increasing it will indeed increase recall, as the model is allowed to predict more classes per instance. However, this can also decrease precision, as the likelihood of predicting incorrect classes also increases.

Given the differing preferences of Top-$K$ and $\lambda$ (with the former favoring recall and the latter favoring precision), we propose an intersection method that considers both parameters for the final CPC classification prediction. This method aims to strike a balance between recall and precision, providing a limited yet reliable set of classification results. By tuning both Top-$K$ and $\lambda$, we can optimize the trade-off between including as many relevant classes as possible (high recall) and minimizing the inclusion of irrelevant classes (high precision).

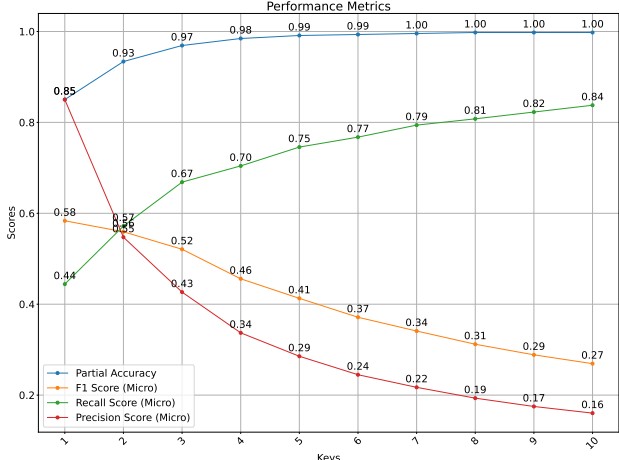

Figure 7: Influence of Top-$K$ inference in CPC classifier.

## F.3 DETAILED RESULTS AND ANALYSIS OF REWRITTEN

As mentioned in Section 4.4, we briefly described the rewriting method; in this section, we present detailed experimental results and analysis.

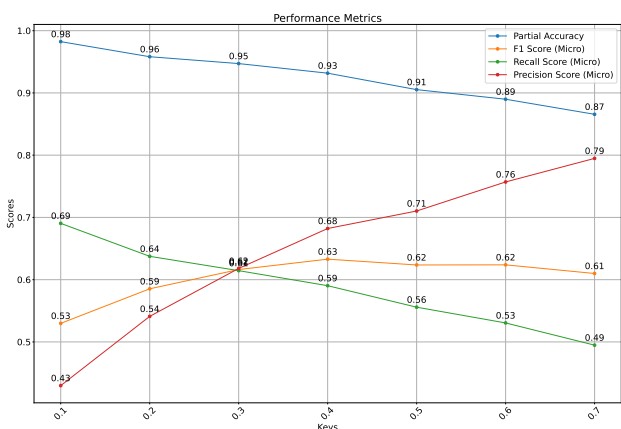

Figure 8: Influence of Thresholding inference in CPC classifier.

Table 14: Matching results of various models on the rewritten A45, A47 and A63 subsets. BGE-large was employed as the encoder model, with Top-500 matching recall (mAR@500) used for the final matching. mPaL refers to the mean length of patent texts, and mPrL refers to the mean length of product texts. The best two mAR@500 and mRoM results are highlighted in red and blue.

| Metric | Subset | Base | Summary | | | Stylistic-Align. |
| | | | Qwen2-0.5B | Qwen2-7B | Llama3-8B | Qwen2-0.5B |
| --- | --- | --- | --- | --- | --- | --- |
| mAR@500↑ | A45 | 14.42 | 27.88 | 14.42 | 31.73 | 8.65 |
| | A47 | 26.81 | 32.61 | 31.88 | 34.06 | 27.54 |
| | A63 | 20.00 | 32.86 | 35.71 | 31.43 | 34.29 |
| mRoM↓ | A45 | 268.38 | 277.68 | 202.61 | 301.77 | 238.60 |
| | A47 | 177.31 | 220.26 | 159.68 | 184.08 | 198.06 |
| | A63 | 125.65 | 189.09 | 206.49 | 240.58 | 148.06 |
| mPaL | A45 | 934.34 | 383.74 | 238.85 | 149.64 | 348.42 |
| | A47 | 995.84 | 391.11 | 244.93 | 151.19 | 357.89 |
| | A63 | 1109.23 | 375.92 | 233.34 | 149.27 | 357.13 |
| mPrL | A45 | 575.67 | 245.36 | 183.97 | 126.24 | 179.67 |
| | A47 | 332.56 | 189.30 | 165.85 | 128.65 | 162.10 |
| | A63 | 329.53 | 169.44 | 175.34 | 132.47 | 173.64 |

**Summarizing Long Contexts to Short Contexts.** In our retrieval task, both patents and product descriptions often contain lengthy sections with extraneous details, making it challenging for an encoder model to focus on the most relevant information. On the other hand, most of SOTA embedding models (e.g., BERT, BGE, T5 and etc) can only handle input text of limited length, which is obviously lower than the length of the original text. Since the part that exceeds the length will be directly truncated, this will obviously cause the loss of information, thus reducing the effectiveness of detection. To address this, we implement a summarization to condense long-form text into shorter, more focused contexts using several LLMs. The goal of the summarization process was to preserve key technical details while eliminating redundant or irrelevant information. This condensed version of the text allows the embedding model to more effectively capture the core semantics of the content, leading to better retrieval accuracy.

We experiment with the A45, A47 and A63 subsets, using Qwen2-0.5B (Yang et al., 2024), Qwen2-7B (Yang et al., 2024), and Llama3-8B (Dubey et al., 2024) instruct models for summarization. To better align these models with our requirements, we used GPT-4o to generate high-quality abstractive summarizing examples, which served as few-shot prompts to guide the models. Details on the use of GPT-4o and querying LLMs can be found in Appendix G. The matching results are shown in Table 14. As shown in the results, all models except Qwen2-7B on the A45 subset outperform the baseline mAR@500 scores, with Qwen2-7B on A45 achieving the same results as the baseline. Additionally, all models significantly reduce the patent and product text lengths (mPaL and mPrL) compared to the original. Furthermore, they exhibit comparable or improved mRoM values,

highlighting the necessity and effectiveness of summarizing patent textual descriptions in helping boost the detection accuracy. Qwen2-0.5B is proved to be significantly faster and more efficient in practical use, achieving a balance between speed and summarization quality, making it the preferred model for this task.

**Rewriting Patent and Product Texts for Similar Stylistic Alignment.**    Another key challenge in product and patent retrieval is the stylistic and linguistic differences between patent texts and product descriptions. Patent documents are often written in a formal, legalistic style, while product descriptions tend to be more commercial and user-friendly. These stylistic discrepancies can create a semantic gap, making it difficult for the embedding model to effectively match similar concepts across the two domains. To address this, we apply a rewriting strategy for stylistic alignment, bridging the semantic gap and enabling the embedding model to better recognize and match concepts across both patent and product description domains.

After determining that Qwen2-0.5B may be optimal for the summarization task, we apply it exclusively for the stylistic alignment task. The matching results are shown in Table 14. By using Qwen2-0.5B for this rewriting task, we observed a significant improvement in matching results for the A63 subset and a slight improvement for the A47 subset. The rewritten texts allowed the embedding model to bridge the semantic gap between patents and product descriptions, resulting in more accurate cosine similarity scores and higher-quality Top-$n$ retrieval results. However, for the A45 subset, we found that the results were lower than the base results. This discrepancy likely arises because, in this case, the models were given either patent texts or product texts individually. The stylistic alignment task heavily relies on the quality of examples provided by GPT-4o, and without strong examples, the models struggled to handle this task effectively using their own knowledge alone. This suggests that the models' intrinsic ability to perform stylistic alignment may need further refinement. Nevertheless, the rewritten method still resulted in shorter mPaL and mPrL values, as well as comparable or improved mRoM scores. While our experiments were conducted on three subsets (A45, A47 and A63), more extensive testing on the entire dataset may require additional resources. We hope that future research will build on these insights and develop more efficient and scalable methods for this task.

## F.4    Detailed Implementations and Analysis About Stretch Matching

While textual information offers a comprehensive description of patents and products, corresponding images often provide more fine-grained and intuitive visual cues, which can serve as crucial supplementary evidence in detecting product-patent infringements. However, as previously mentioned, the significant domain gap between patent and product images makes direct similarity measurement highly challenging. To address this issue, in our work, we propose a simple yet effective style-transfer method based on stretch detection to alleviate the domain shift, and then utilize the powerful CLIP model to extract unified feature representations. Specifically, as shown in Figure 3, we transform the original patent and product images, which exhibit substantial domain disparities, into a similar visual style using stretch detection (Zhou et al., 2024). Subsequently, CLIP is employed to extract feature embeddings for each image. For instances with multiple associated images, we compute the average of their embeddings to form a unified representation. By calculating the cosine similarity between feature representations, we retrieve potentially infringing patents for each product. Results in Table 9 demonstrate that, benefiting from stretch-based style transfer, we achieve a mAR@500 of 33.92%, marking a 21.42% improvement over using raw natural images. Considering the significant performance and low computational cost of text-based retrieval, we further propose a hierarchical text-to-image retrieval strategy. In the first stage, we utilize text matching to filter the Top-$h$ most likely patent candidates for each product, thus narrowing the retrieval pool from a vast, shared collection to a tailored, more-focused subset, which not only reduces computational complexity but also simplifies the matching process. Next, we perform image-based retrieval within the selected Top-$h$ candidates. With this approach, we further enhance image-based retrieval performance, increasing mAR@500 from 33.92% to 42.85%. $h$ is set to 5000 in our experiments, as the mAR@5000 for text-based retrieval achieves approximately 97%, ensuring the inclusion of almost all potentially infringing patents.

## F.5 DETAILS OF ALTERNATIVE METRICS TO COSINE SIMILARITY

In this section, we compare the performance of the SFT-based metric against the standard cosine similarity approach. As discussed in Section SFT, cosine similarity may have limitations in capturing deeper semantic relevance between product and patent embeddings. To address this, we explore two variants of the SFT-based method:

- **SFT Classifier Only**: In this setup, we use the pretrained embeddings from the encoder and apply SFT to train a binary classifier based on the product-patent pairs. The classifier's output logits are used as the metric for measuring semantic relevance.

- **SFT Encoder and Classifier**: This approach trains the entire model, including both the encoder and the classifier, in an end-to-end fashion during fine-tuning. By jointly optimizing both components, this paradigm allows the model to learn more task-specific representations of product and patent embeddings. The output logits from the classifier serve as the metric for semantic relevance between paired products and patents.

Table 15: Matching results comparing cosine similarity and SFT-based approaches (SFT Classifier Only and SFT Classifier & Encoder). mAR@500 are reported across various subsets and the best "ALL" results for BGE-large (vanilla) and BGE-large (fine-tuned) are highlighted separately.

| Method | | ALL | A45 | A47 | A63 | B65 | H01 |
|---|---|---|---|---|---|---|---|
| BGE-large (vanilla) | Cosine. | 16.08 | 14.42 | 26.81 | 20.00 | 18.54 | 29.01 |
| | SFT Cls. Only | **48.02** | 95.19 | 61.59 | 50.00 | 50.00 | 29.75 |
| | SFT Cls. & Enc. | 27.97 | 88.46 | 50.00 | 58.57 | 39.02 | 40.50 |
| BGE-large (finetuned) | Cosine. | **65.32** | 75.00 | 73.19 | 62.86 | 75.61 | 59.60 |
| | SFT Cls. Only | 52.86 | 82.69 | 70.29 | 47.14 | 57.32 | 46.28 |
| | SFT Cls. & Enc. | 37.22 | 96.15 | 68.84 | 60.00 | 45.12 | 16.53 |

From the results presented in Table 15, several key observations can be made regarding the performance of the different methods:

- **SFT Classifier Only vs. SFT Classifier and Encoder**: Generally, the SFT Classifier Only approach outperforms the SFT Classifier and Encoder across most subsets. This suggests that fine-tuning the classifier alone yields better semantic matching performance, while involving the encoder in the fine-tuning process may hinder the model's ability to capture useful semantic information. One possible explanation is that the SFT objective, which optimizes the model for a binary classification task, may interfere with the encoder's original capacity to represent semantic relationships. By focusing on this specific task, the encoder might lose some of its generalizability, leading to a reduction in the ability to capture more nuanced semantic features.

- **Vanilla BGE v.s. Fine-tuned BGE**: Another notable observation is the comparison between the vanilla and fine-tuned BGE models. For the vanilla BGE, the SFT-based approach (especially the SFT Classifier Only) performs significantly better than cosine similarity. This indicates that applying SFT enhances the model's ability to identify relevant product-patent pairs. However, in the case of fine-tuned BGE, cosine similarity performs better than the SFT-based metrics. This could be because the fine-tuned BGE model has already been optimized for specific task-related semantic matching during its fine-tuning process. In this case, the simple cosine similarity metric might be more effective in capturing the representations learned by the model, whereas the SFT-based approach introduces additional complexity that may not be necessary or helpful after the encoder has already undergone task-specific fine-tuning.

Based on these observations, several potential directions can be explored:

- **Designing a more complex classifier network**: The current binary classifier could be enhanced by introducing more sophisticated architectures, such as deeper neural networks or attention-based mechanisms, to better capture semantic relevance between product and patent pairs.

- **Finding a better loss function**: The current loss function used in the SFT training might not fully optimize the model's ability to differentiate between semantically relevant and irrelevant pairs. Exploring alternative loss functions, such as contrastive loss or triplet loss, could help improve the model's ability to measure semantic relevance more effectively.

- **Exploring better objectives to replace cosine similarity**: Since cosine similarity may not always capture the full complexity of semantic relationships, particularly after fine-tuning, it might be beneficial to investigate alternative metrics or objectives for evaluating the relevance between embeddings. This could include metrics that account for the contextual nuances of product and patent texts.

### F.6 VISUAL-ENHANCED MULTI-MODALITY INFRINGEMENT DETECTION

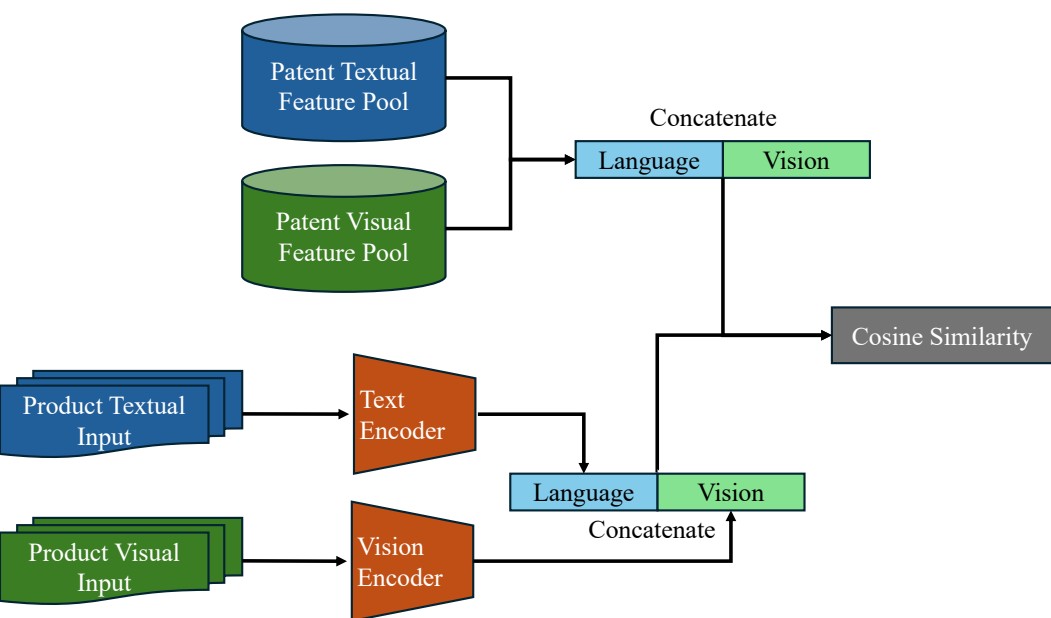

Figure 9: Our visual-enhanced multi-modality infringement detection framework.

In this section, we delve into an effective analysis on the fusion of textual and visual features to achieve efficient infringement detection. The integration of these two types of features has been a challenging task due to their inherently distinct natures. However, our study indicates that they can complement each other in the context of patent infringement detection, potentially leading to improved results over using either modality alone.

Textual features provide a detailed account of the patented technology. They capture the nuances of the technology's functionality, design, and implementation. However, they may not effectively represent the visual aspects of the technology, such as its physical design, color, or shape, which are often crucial in determining infringement. On the other hand, visual features, extracted from patent drawings or product images, can capture these visual aspects. They can effectively represent the physical appearance of the technology, which can be crucial in some infringement cases. However, they may not capture the functional or implementation details that are often described textually.

Therefore, a comprehensive infringement detection system should ideally incorporate both textual and visual features. As shown in Figure 9, we design a multi-modality infringement detection framework based on concatenation between visual features and textual features.

We conduct experiments in A63 subset and the experimental results are summarized in Table 16, from which we obtain several key insights and analysis:

- **Performance of Pure Modalities:** The pure text modality, without any rewriting, significantly outperforms the image-based approach in terms of mAR@500. This indicates that

Table 16: Comparative performance of different infringement detection methods. The table shows the mAR@500 and mRoM for various methods, including pure text, pure image, simple concatenation of the two, and more sophisticated fusion and voting methods. The results highlight the potential of multi-modal fusion and voting for improving infringement detection.

| Method | mAR@500 | mRoM |
|---|---|---|
| Pure Text Baseline | 71.43 | 235.34 |
| Pure Image Baseline | 57.14 | 71.32 |
| Concatenation | 69.64 | 144.46 |
| Concatenation **Should** | 73.78 | - |
| Text-to-Image and then Vote **Should** | 87.59 | - |

textual data provides a more comprehensive and detailed description of the patent, allowing for more accurate detection of infringements. However, the image modality has a higher mRoM, meaning that the patents it does detect as infringements are more likely to be truly infringing. This suggests that images can capture certain aspects of patents that text may miss, making them valuable for infringement detection.

- **Concatenation of Modalities:** A simple concatenation of the two modalities does not yield optimal results. While it does perform better than the image baseline, it falls short of the performance of the pure text approach. This suggests that a naive fusion of the two modalities is not sufficient to fully leverage their complementary strengths.

- **Potential of Fusion:** Our analysis reveals that images can detect some infringing patents that text fails to identify. Ideally, if these unique detections could be perfectly combined, the mAR@500 could reach 73.78. This underscores the need for a more sophisticated fusion method that can effectively harness the complementary aspects of the two modalities. It also reaffirms that the perfect infringement detector would rely on both image and text modalities.

- **Further Improvement with Voting:** When we further apply our proposed text-to-image hierarchical detection method and compare the results of image and text detections, we find that even more infringing samples can be identified, with the mAR@500 potentially reaching 87.59. This highlights two key points: firstly, our text-to-image hierarchical detection method is effective; and secondly, applying a voting-style post-processing to the results of different modalities can yield improved detection results.

In summary, these results stress the importance of developing a more sophisticated method for fusing text and image modalities and applying a suitable voting mechanism to balance their contributions and demonstrates that such a multi-modal approach can indeed lead to better infringement detection results. However, our study also reveals that simply combining these features in a naive way, such as by concatenation or averaging, does not achieve optimal results. This leads us to the conclusion that a more sophisticated fusion or voting method is needed to balance the prominence of visual and textual features. This method should be capable of weighing the contributions of each modality according to its relevance to the specific infringement case at hand.

## G PROMPTS FOR UTILIZING GPT-4O AND QUERYING LLMS

Recall in Section 4.4 and Appendix F.3, we used GPT-4o to generate high-quality summarization examples. Below is the specific template used to query GPT-4o.

**For summarization:**

### System Prompt: *You are a helpful assistant to help summarize the context of a patent abstract along with the corresponding claim. Ensure that the summary of patent contexts captures the essence of both the abstract and the claim.*
### User: *Patent Abstract: {Abstract}*
*Patent Claim: {Claim}*
### Assistant:

---

### System Prompt: *You are a helpful assistant to help summarize the context of a product description.*
### User: *Product Description: {Desc}*
### Assistant:

**For stylistic alignment:**

### System Prompt: *You are a helpful assistant to help summarize the context of a patent abstract along with the corresponding claim or a product description. Ensure that the summary of patent contexts captures the essence of both the abstract and the claim. Furthermore, ensure the summarization of patent contexts and product contexts have the same style. In each communication, you will receive either the patent contexts or product contexts separately.*
### User: *Patent Abstract: {Abstract}*
*Patent Claim: {Claim}*
### Assistant:

---

### System Prompt: *You are a helpful assistant to help summarize the context of a patent abstract along with the corresponding claim or a product description. Ensure that the summary of patent contexts captures the essence of both the abstract and the claim. Furthermore, ensure the summarization of patent contexts and product contexts have the same style. In each communication, you will receive either the patent contexts or product contexts separately.*
### User: *Product Description: {Desc}*
### Assistant:

After obtaining examples from GPT-4o, these examples serve as few-shot prompts. Below is an example of summarizing a patent (the structure is similar for others):

### System Prompt: *You are a helpful assistant to help summarize the context of a patent abstract along with the corresponding claim. Ensure that the summary of patent contexts captures the essence of both the abstract and the claim. I will give you some examples as follows.*
*Example 1:*
*Patent Abstract: {Abstract}*
*Patent Claim: {Abstract}*
*Summary: {Summary from GPT-4o}*
*Example 2:*
*Patent Abstract: {Abstract}*
*Patent Claim: {Abstract}*
*Summary: {Summary from GPT-4o}*
### User: *Patent Abstract: {Abstract}*
*Patent Claim: {Abstract}*
### Assistant:

| | |
|---|---|
| **Patent Abstract** | An apparatus for dispensing a liquid is disclosed. The apparatus can include a cap configured to connect to a container body having a chamber for containing the liquid. The cap can include a fill aperture through which the liquid is supplied to the container body and a pour aperture through which the liquid exits the apparatus. A fill lid can be rotatable about a pivot axis in a first direction to close the fill aperture and rotatable about the pivot axis in a second direction to open the fill aperture the first direction opposite the second direction. A pour lid can be rotatable about the pivot axis in the first direction to open the pour aperture and rotatable about the pivot axis in the second direction to close the pour aperture. A filter assembly comprising a filter cartridge can connect to the cap. |
| **Patent Claim** | An apparatus for dispensing a liquid the apparatus comprising a container body a cap configured to removably attach to the container body and having a chamber for containing the liquid the cap comprising a first vent through the cap to enable air to pass from the chamber of the container body through the first vent and outside the container body a fill aperture through which the liquid is supplied to the container body a pour aperture through which the liquid exits the apparatus a fill lid rotatable about a pivot axis in a first direction to close the fill aperture and rotatable about the pivot axis in a second direction to open the fill aperture the first direction opposite the second direction the pivot axis nonparallel to a longitudinal axis of the apparatus and a pour lid rotatable about the pivot axis in the first direction to open the pour aperture and rotatable about the pivot axis in the second direction to close the pour aperture and a filter assembly which connects to the cap the filter assembly comprising a filter cavity to contain filtration media. The apparatus of claim wherein the filter assembly comprises a filter body with a top filter pad coupled with an inlet of the filter assembly and a bottom filter pad coupled with an outlet of the filter assembly. The apparatus of claim wherein the filter assembly further comprises a sleeve comprising a first opening a second opening and an annular wall extending between the first opening and the second opening to define the filter cavity the filter assembly comprising a filter cartridge disposed in the filter cavity. The apparatus of claim wherein the filter assembly comprises a mesh basket comprising a top mesh filter pad a bottom mesh filter pad and a mesh wall extending between the top mesh filter pad and the bottom mesh filter pad the mesh basket at least partially defining the filter cavity. The apparatus of claim wherein the mesh basket comprises stainless steel. The apparatus of claim wherein the filter assembly further comprises a verticallyextending flange disposed about the perimeter of an outer surface of the top mesh filter pad. The apparatus of claim wherein a pore size of the top mesh filter pad is larger than a pore size of the bottom mesh filter pad. The apparatus of claim wherein the cap comprises a platform disposed below the pour lid and the fill lid and a sidewall extending transversely from a bottom side of the platform the sidewall defining a recess sized and shaped to receive the filter assembly the sidewall and recess configured to be disposed within the chamber of the container body. The apparatus of claim wherein the first vent is disposed through the sidewall to provide fluid communication between the chamber of the container body and the recess. The apparatus of claim further comprising a second vent through the platform to provide fluid communication between the recess and the outside environs. The apparatus of claim wherein the cap has a pouring configuration in which the liquid exits the apparatus and a filling configuration in which the liquid is supplied to the container body wherein when the cap is in the filling configuration the second vent is occluded and wherein when the cap is in the pouring configuration the second vent is open. The apparatus of claim further comprising the filter assembly coupled with the sidewall and disposed at least partially in the recess the filter assembly having an outwardlyextending flange at an upper end portion of the filter assembly the flange disposed in a corresponding groove of the sidewall. The apparatus of claim further comprising a diffuser disposed above the filter assembly the diffuser comprising a diffuser body having one or a plurality of diffuser openings therethrough. An apparatus for dispensing a liquid the apparatus comprising a cap configured to connect to a container body having a chamber for containing the liquid the cap comprising a first vent through the cap to enable air to pass from the chamber of the container body through the first vent and outside the container body a fill aperture through which the liquid is supplied to the container body a pour aperture through which the liquid exits the apparatus a fill lid rotatable about a pivot axis in a first direction to close the fill aperture and rotatable about the pivot axis in a second direction to open the fill aperture the first direction opposite the second direction the pivot axis nonparallel to a longitudinal axis of the apparatus a pour lid rotatable about the pivot axis in the first direction to open the pour aperture and rotatable about the pivot axis in the second direction to close the pour aperture and a filter assembly which connects to the cap the filter assembly comprising a filter cavity to contain filtration media wherein the filtration media is disposed within the filter assembly. The apparatus of claim wherein the filtration media comprises granular activated carbon. |
| **Summary** | The patent describes an apparatus for dispensing a liquid, which includes a cap that connects to a container body and has a chamber for containing the liquid. The cap has a fill aperture for supplying the liquid to the container body and a pour aperture for dispensing the liquid. The cap also includes a fill lid and a pour lid that are rotatable about a pivot axis to open and close the respective apertures. A filter assembly, which contains filtration media, is connected to the cap and is designed to filter the liquid before it is dispensed. The apparatus also includes a first vent to allow air to pass from the chamber to the outside and a second vent that is occluded when the cap is in the filling configuration and open when it is in the pouring configuration. The filter assembly can be a mesh basket or a filter cartridge, and the filtration media can be granular activated carbon. |

Figure 10: Example of patent text summarization.

# H EXAMPLES OF REWRITTEN

Figures 10 through 12 provide examples demonstrating the effectiveness of the summarization and stylistic alignment methods. Both patent and product texts often contain redundant information, and the summarization process significantly reduces their length while preserving key content.

| Product Title | 1x Golf Weight Fit for Callaway Epic Flash Sub Zero Driver 2019 |
|---|---|
| Product Description | Product Description Golf Weight Fit for Callaway Epic Flash Sub Zero SZ Driver 2019 Package include: 1 piece weight(the driver clubs isn't include)\n\nProduct information Technical Details Brand Name \u200eDree Manufacturer \u200eDree Hand Orientation \u200eRight Additional Information ASIN B07T4265FR Customer Reviews 3.1 3.1 out of 5 stars 8 ratings 3.1 out of 5 stars Best Sellers Rank #1,495,081 in Sports & Outdoors (See Top 100 in Sports & Outdoors) #5,070 in Golf Training Equipment #8,302 in Golf Accessories Date First Available June 17, 2019 Feedback Would you like to tell us about a lower price? 1x Golf Weight Fit for Callaway Epic Flash Sub Zero Driver 2019 Share: Found a lower price? Let us know. Although we can't match every price reported, we'll use your feedback to ensure that our prices remain competitive. Where did you see a lower price? Website (Online) URL: Price: ($) Shipping cost: ($) Date of the price: 01 02 03 04 05 06 07 08 09 10 11 12 / 01 02 03 04 05 06 07 08 09 10 11 12 13 14 15 16 17 18 19 20 21 22 23 24 25 26 27 28 29 30 31 / Store (Offline) Store name: City: State: Please select province Please select province Price: ($) Date of the price: 01 02 03 04 05 06 07 08 09 10 11 12 / 01 02 03 04 05 06 07 08 09 10 11 12 13 14 15 16 17 18 19 20 21 22 23 24 25 26 27 28 29 30 31 / Please sign in to provide feedback. Submit Feedback\n |
| Summary | The product is a golf weight designed specifically for the Callaway Epic Flash Sub Zero SZ Driver 2019. It is a single piece weight that does not include the driver clubs. The product is manufactured by Dree and is suitable for right-handed players. As of its first availability on June 17, 2019, the product has received a customer rating of 3.1 out of 5 stars based on 8 ratings. The product ranks #1,495,081 in Sports & Outdoors, #5,070 in Golf Training Equipment, and #8,302 in Golf Accessories. The seller encourages feedback and is open to price matching if customers find a lower price elsewhere. |

Figure 11: Example of product text summarization.

| Patent Abstract | An exercise training device for use with a motorcycle for working out to improve a riders physical fitness is provided. The training device includes a base a mounting plate and a plurality of support legs extending from the base. A coil spring is vertically mounted between the base and the mounting plate. A motorcycle can be releasably affixed to the mounting plate with a plurality of hold downs. The coil spring deflects in response to lateral forces such that the affixed motorcycle can pitch and roll when subject to sufficient force. |
|---|---|
| Patent Claim | An exercise training device for use with a motorcycle the training device comprising a base having a central surface a mounting plate having an upper surface and a lower surface a coil spring having a longitudinal axis and vertically mounted between said base and said mounting plate an upper end of said coil spring attached to said lower surface of said mounting plate a lower end of said coil spring attached to said central surface of said base a plurality of hold downs configured to releasably affix a motorcycle to said mounting plate and a pair of restrictors attached to said base and said mounting plate wherein said coil spring deflects in response to forces applied laterally to the longitudinal axis of said coil spring such that said affixed motorcycle can pitch and roll when subject to sufficient force. The exercise training device of claim wherein a rider seated on said motorcycle can controllably maneuver said motorcycle between an initial vertical position and an inclined position where said motorcycle is inclined to the left or right of said vertical position. The exercise training device of claim wherein said rider maneuvers the position of said motorcycle by leaning to one side or the other. The exercise training device of claim wherein a left restrictor restricts the degree of incline to which said motorcycle can be maneuvered to the right and a right restrictor restricts the degree of incline to which said motorcycle can be maneuvered to the left. The exercise training device of claim wherein said base includes elongated legs that extend out from said base. The exercise training device of claim wherein said legs each include a hole at a distal end thereof for affixing said training device to a floor surface. The exercise training device of claim wherein said legs each include a height adjustable foot at a distal end thereof. The exercise training device of claim wherein an underside of said base includes reinforcing structural members. The exercise training device of claim wherein said mounting plate includes a plurality of slots therethrough. The exercise training device of claim wherein said hold downs extend through said slots in said mounting plate and the position of said hold down within said slot may be adjusted to accommodate the structures of different motorcycles. The exercise training device of claim wherein said base includes a plurality of through holes configured for fasteners to extend therethrough for affixing said exercise training device to a floor surface. The exercise training device of claim wherein said coil spring defines upper and lower ends the upper end can pivot substantially about said lower end of said coil spring affixed to said base. The exercise training device of claim wherein said restrictors restrict how far said motorcycle mounted to said mounting plate can pivot. An exercise training device for developing skills useful for motocross racing comprising a base a biasing element extending upwardly from said base to a mount with said mount being spaced from said base by said biasing member and said mount being configured to secure a frame of a motorcycle to said biasing element said biasing element being compressible in a vertical direction and providing three hundred sixty degrees of movement in a direction offset from the vertical direction being representative of movement of a motorcycle while motocross racing and opposing restrictors for restricting movement in a lateral direction relative to said frame of the motorcycle. The training device of claim wherein said biasing member comprises a coil spring. The training device of claim wherein said opposing restrictors comprise opposing straps extending between said mount and said base thereby. The training device of claim wherein said mount includes a plurality of hold downs each defining a releasable hook for releasably securing the frame of the motorcycle to said mount. The training device of claim wherein said biasing member retains the frame of the motorcycle in an upright position and allows pitch and roll of the frame of the motorcycle when subject to force sufficient to overcome the bias of said biasing member. The training device of claim wherein said opposing restrictors prevent movement in a lateral direction relative to said frame of the motorcycle beyond about forty five degrees from the vertical orientation. |
| Patent Summary | This patent describes an exercise training device for use with a motorcycle for improving physical fitness. The device includes a base, mounting plate, and support legs extending from the base. A coil spring is vertically mounted between the base and the mounting plate. A motorcycle can be releasably affixed to the mounting plate with a plurality of hold downs. The coil spring deflects in response to lateral forces such that the affixed motorcycle can pitch and roll when subject to sufficient force. The exercise training device of claim is suitable for use with a motorcycle for improving physical fitness. |
| Product Title | Balanced Body Fletcher Pilates Towel, Workout Towel for Pilates Equipment, Exercise Accessory and Prop for Shoulder, Arm, and Chest Workouts |
| Product Description | Product Description Balanced Body Fletcher Pilates Towel, Workout Towel for Pilates Equipment, Exercise Accessory and Prop for Shoulder, Arm, and Chest Workouts\n\nProduct information Brand Balanced Body Towel form type Hand Towel Age Range (Description) All Ages Material Cotton Number of Items 1 Special Feature Non slip Theme No Theme Product Care Instructions Machine Wash Size Small Unit Count 1.0 Count Fabric Type 100% Cotton Item Weight 1.2 pounds ASIN B004AWX236 Customer Reviews 4.6 4.6 out of 5 stars 30 ratings 4.6 out of 5 stars Best Sellers Rank #155,668 in Sports & Outdoors (See Top 100 in Sports & Outdoors) #209 in Pilates Equipment Is Discontinued By Manufacturer No Date First Available November 5, 2010 Feedback Would you like to tell us about a lower price? Balanced Body Fletcher Pilates Towel, Workout Towel for Pilates Equipment, Exercise Accessory and Prop for Shoulder, Arm, and Chest Workouts Share: Found a lower price? Let us know. Although we can't match every price reported, we'll use your feedback to ensure that our prices remain competitive. Where did you see a lower price? Website (Online) URL: Price: ($) Shipping cost: ($) Date of the price: 01 02 03 04 05 06 07 08 09 10 11 12 / 01 02 03 04 05 06 07 08 09 10 11 12 13 14 15 16 17 18 19 20 21 22 23 24 25 26 27 28 29 30 31 / Store (Offline) Store name: City: State: Please select province Please select province Price: ($) Date of the price: 01 02 03 04 05 06 07 08 09 10 11 12 / 01 02 03 04 05 06 07 08 09 10 11 12 13 14 15 16 17 18 19 20 21 22 23 24 25 26 27 28 29 30 31 / Please sign in to provide feedback. Submit Feedback\nFrom the brand Previous page BALANCED BODY REFORMERS Some of the most versatile pieces of Pilates equipment available, our Reformers are perfect for your home gym, club, or Pilates studio! Pilates Reformers Visit the Store Customer Favorites Visit the Store STRENGTH-TRAINING EQUIPMENT Balanced Body Pilates equipment is ideal for building both strength and flexibility while training the whole body. Strength & Flexibility Tools Visit the Store Pilates Chairs Visit the Store PILATES ACCESSORIES Balanced Body offers an array of Pilates workout tools and accessories to challenge and support your needs at home or in the studio. Training Tools Visit the Store Reformers with Towers and Mats Visit the Store Next page |
| Product Summary | The product description is a Pilates Towel, Workout Towel for Pilates Equipment, Exercise Accessory and Prop for Shoulder, Arm, and Chest Workouts. The product is a hand towel that is made of cotton and has a non-slip theme. It is available in different sizes and comes with a limited lifetime warranty. |

Figure 12: Example of summarizing and aligning patent and product texts to maintain consistency in style.

