# OpenReview forum: "ERiC-UP$^3$ Benchmark: E-Commerce Risk Intelligence Classifier for Detecting Infringements Based on Utility Patent and Product Pairs"
_ICLR.cc/2025/Conference — Submitted to ICLR 2025_

### Official Review · Reviewer_Pmou · 2024-10-31

**Soundness:** 2
**Presentation:** 2
**Contribution:** 2
**Rating:** 3
**Confidence:** 5

**Summary:**

This paper presents ERiC-UP$^3$, a dataset with annotations to detect infringement behaviors between a given Amazon product and existing patents. This dataset includes 1 million product samples, 13 million patent samples, and 11,000 meticulously annotated infringement pairs for training and 2,000 for testing. This work benchmarks existing baselines and proposes a two-stage pipeline for effectively conducting infringement detection. This paper also provides some best practices to improve detection.

**Strengths:**

1. A new dataset for detecting infringements in the patent domain (the unique aspect lies in the annotation).
2. A proposed pipeline to surpass existing methods.
3. Some useful takeaways to improve detection.

**Weaknesses:**

1. The dataset offers some novelty but is largely a domain adaptation from existing datasets like [1] and [2]. Its main advantage lies in expert annotations on infringement cases. However, the dataset’s scale is limited, with relatively few annotations, and the patent and product samples were scraped from the internet. Additionally, the distinction between "base" and "large" versions is minimal.

2. The writing lacks clarity, making it hard to grasp key points at first glance: (1) Is infringement treated as a ranking problem? (2) What constitutes the "domain gap"? Is it simply a stylistic shift? (3) Why were these particular classes selected?

3. The technical pipeline appears ad hoc. Why use a two-stage approach instead of a streamlined, end-to-end model? Why can't existing models address this problem effectively? Why wasn’t the current infringement detection pipeline integrated into the study?

4. Key baselines are missing from this study: (1) multimodal baselines, such as LLaVA, and (2) baselines from prior infringement detection research.

5. Important ablations on the pipeline components are absent. For instance, how does removing expert labels affect training? What are the results if detections are run without training labels?

6. The analysis part is shallow, with findings that are largely known within the field.

7. The literature review lacks the recent works.

8. Some obvious typos in number and upper/lower case.

[1] A Dataset and Benchmark for Copyright Infringement Unlearning from Text-to-Image Diffusion Models
[2] TMID: A Comprehensive Real-world Dataset for Trademark Infringement Detection in E-Commerce

**Questions:**

see wearkness. I also has a question about the significant of this work: 1) Can Google Patents (https://patents.google.com/) be used for detect infringement? 2) Is Amazon conduct infringement screening before releasing the product? If they do so, i think that only very limited samples in the ERiC-UP$^3$ involves infringement, and training model with ERiC-UP$^3$ cannot significant detect real-world product.

---

### Official Review · Reviewer_yHKi · 2024-11-04

**Soundness:** 2
**Presentation:** 2
**Contribution:** 2
**Rating:** 5
**Confidence:** 4

**Summary:**

The article proposes a new task of detecting potential infringing patents for a given product, and introduces a large-scale MultiModal Machine Learning dataset called ERiC-UP3, aimed at promoting research in this field. The dataset contains over 13 million patent samples and 1 million product samples, providing real-world scenarios needed for deep functional understanding to promote innovative and practical solutions for intellectual property infringement detection. It also provides some evaluation baselines and testing methods. In essence, it has the following setting:

Search task set: Retrieve the patent q that product p is most likely to infringe and give the probability ranking of infringing patents in the patent list

Task objective: Ensure that patents with the most similar functions and potential infringement are ranked highest in the list sorting

**Strengths:**

Marking products and infringing data is a very heavy workload, this work has done some valuable efforts to annotate the data.

The text data of the product and the patent have been rewritten, which can avoid the potential meaning difference.

**Weaknesses:**

The writing is rather confused.

This article mainly discloses the patent product infringement data of MultiModal Machine Learning, but the main text mainly discusses the single modality of text, and there is little use and verification of image modality.
For datasets, some graphic and table information is invalid. There is a lot of redundant information in the paired graphic and text dataset. This article rarely mentions and verifies how to ensure that this data is effective for training, and rarely uses this data for experiments and verification of graphic and text information for infringement conflicts.
The expert evaluation mentioned in the article mainly evaluates whether there is infringement between the product and the patent, rather than evaluating the validity of the data

However, from the perspective of CS, it is not clear whether these MultiModal Machine Learning data are effective and what the purpose of using these data is.


I have several questions regarding this work:

1. The overall framework of the paper is quite chaotic, and the research framework is not clear.

2. The experiment is comprehensive, but many tables have unclear meanings and are chaotic, a bit like an experimental report
  - The Table 7 compares which method, I can't tell, and it's not specifically written in the article, just said the score is high.
  - In Table 8, Using LLM to rewrite the text data of the product and the patent can effectively avoid the difference in emphasis between the two. However, it is hasty and inaccurate to determine that 0.5b qwen is the best for only three categories. Llama3-8b also has multiple high scores. Why not consider llama3-8b?


3. In Figure 6, the significance of calculating the recall rate of the top 500 is not great, and the average value of each CPC category is not given, and it cannot be seen that this mAR@500 is a good evaluation index, and the number of samples of each CPC classification is very different, the variance is very large, why not use top 10% or top 1% as the evaluation index, as shown in the figure below is the order of magnitude of 10 ^ 6.

4. The experimental framework of MultiModal Machine Learning fusion retrieval is not clear
  For example, how to evaluate after image retrieval mAR@500 scores are not given
  Why is it first evaluated through text matching to the relevant patent pool, and then evaluated through image retrieval, rather than directly conducting image retrieval (missing this experiment)?

  This article mainly conducts experiments on text modality, with little emphasis on the role of MultiModal Machine Learning data, and does not reflect the significance of MultiModal Machine Learning data for infringement retrieval.

Regarding the experiment of MultiModal Machine Learning in this article, the following questions are raised:

- The MultiModal Machine Learning experiment in the main text of this article is just a simple stitching, text classification + image retrieval. Where is the specific integration of MultiModal Machine Learning reflected?
- The experiments are all here and there, and the overall performance of MultiModal Machine Learning cannot be seen
- Table 9 shows the image retrieval results after text classification. Which method is used for the first step of text classification? Or is it directly given classification for image retrieval to eliminate errors caused by text classification? If not, how to eliminate errors? Why not directly retrieve images? The explanation is not comprehensive enough.


5. The experimental data is incorrect. The experimental table of MultiModal Machine Learning in the above text is different from the data given in the last supplementary material.
    - The experimental data of the plain text of table5 and table16 are inconsistent, and there is no other data of 71.43.

6. Many of the experiments in the supplementary materials are not mentioned in the main text, and there is no clear definition of how the methods are done, how to conduct mixed experiments, or how to conduct mixed voting, and how to evaluate them
   - According to Table 16, the article only mentions simple concatenation, but does not explain how the following two fusion are done, and the description is quite confusing. Is the voting experiment at the end just a simple union of the results of the baselines of the original two modes? Or are there other voting operations?

**Questions:**

as above

---

### Official Review · Reviewer_Jxy9 · 2024-11-07

**Soundness:** 3
**Presentation:** 2
**Contribution:** 2
**Rating:** 5
**Confidence:** 3

**Summary:**

This paper proposes a benchmark dataset for E-commerce intelligence for machine learning field. This research narrows the gap between the E-commerce area with current artificial intelligence research. In addition, this draft further provides analysis for the proposed benchmark and gives several baseline methods for reference to following research works.

**Strengths:**

1. This work narrows the gap between e-commerce and machine learning research, which is a valuable try and has potential to further enlarge the impact of machine learing.
2. In addition to the proposed benchmark dataset, it also provides detailed statistical analysis with several backbone experiments for reference.
3. Overall, the writing is easy to follow.

**Weaknesses:**

1. Some parts of the draft are not well-prepared, such as tab.7 and 8. The overall format needs a careful polish.
2. Even if the proposed benchmark is for multi-modal learning, especially for vision-language interaction, I still think this topic fits better for data mining or multi-media conferences, especially considering it is a dataset-oriented paper.
3. Captions of figures and tables are necessary to be enriched. At least, they need to indicate the conclusion of the tables and figures. Overall, they need to be more informative.

**Questions:**

Please check the weaknesses section above.

---

### Meta-Review · Area_Chair_gAo1 · 2024-12-18

**Metareview:**

This paper studies the problem of detecting potential infringing patent for a given product. To achieve it, the authors first introduce a large-scale multimodal dataset called ERiC-UP3, then a two-stage infringement detection solution is proposed.

The reviewers all acknowledge the importance of the newly proposed dataset, and recognize the heavy workload of the annotation. Some reviewers also appreciate the proposed pipeline and recognize the success of suppress other baselines in the evaluation. While a number of concerns and questions are raised. 1. The paper structure and writing are unclear and not well organized,  particularly the figures are not well explained pointed by the reviewers. 2. The experiments focus mostly on text, lacking proper validation of the image modality of the multimodal data. 3. The dataset has some novelty with expert annotations, but is limited in scale.  Hence, I conclude that the paper could not be accepted in its current form and would require a major revision.

**Additional Comments On Reviewer Discussion:**

No rebuttal provided.

---

### Decision · Program_Chairs · 2025-01-22

Reject